# Kynurenine mediates the chemotherapy-induced intestinal toxicity through modulation of gut microbiota

Hongyu Xie[1,2,7], Jingyi Yang[2,3,4,5,7], Jinjie Wu[2,3,4,5,7], Wenhao Ma[2,3,4,5], Haoyang Xu[2,3,4,5], Shuang Guo[2,4,5], Yanchun Xie[2,4,5], Zhanhao Luo[2,4,5], Dayi Liang[2,3,4,5], Mujia Cao[2,3,4,5], Danling Liu[3,4,5], Sanqing Jin [1] ✉, Ping Lan [2,3,4,6] ✉ & Zhen He [2,3,4,5] ✉

Chemotherapy-induced intestinal toxicity is a major dose-limiting complication, but the underlying mechanisms linking systemic metabolism to localized gut damage are poorly understood. Here we show that serum L-kynurenine, a tryptophan metabolite, is elevated in patients with severe oxaliplatin-induced intestinal toxicity. Accumulation of L-kynurenine is driven by IFNγ-mediated induction of indoleamine 2,3-dioxygenase 1 (IDO1) in myeloid cells. Using scRNA-seq and myeloid cell-specific knockout models, we confirm that myeloid cell-derived L-kynurenine exacerbates toxicity. Critically, L-kynurenine accumulation drives gut dysbiosis, characterized by the loss of *Lactobacillus johnsonii*, and subsequently activates the TNFα/JNK pathway, leading to intestinal epithelial apoptosis. Pharmacological inhibition or engineered reduction of L-kynurenine mitigates chemotherapy-induced intestinal injury. Our findings reveal an important role of L-kynurenine from myeloid cells in chemotherapy tolerance and propose its targeting as a potential therapeutic strategy.

Cytotoxic chemotherapeutics are crucial for cancer treatment because they primarily kill tumor cells[1]. However, the occurrence of chemotherapy-induced side effects restricts the ability of cancer patients to maintain appropriate therapeutic dosing and duration, in turn limiting treatment efficacy[2]. One of the most common side effects of these drugs, affecting 40–80% of patients, is the onset of gastrointestinal toxicity[3]. Oxaliplatin, one of the common drugs used to treat colorectal cancer (CRC), is associated with neural and intestinal toxicity[4]. Therefore, a better understanding of intestinal toxicity in response to oxaliplatin is desperately needed, so that new concepts can be developed for future therapies.

Metabolic reprogramming plays a critical role in cancer therapy[5]. An increasing number of studies have further emphasized the significance of tryptophan metabolism in the chemotherapeutic response. A higher level of 3-indoleacetic acid (3-IAA) in serum has been demonstrated to be associated with increased treatment sensitivity in pancreatic cancer[6]. Indole-3-propionic acid (IPA) has also been shown to improve immune checkpoint blockade responsiveness by modulating the stemness program of CD8+T cells[7]. In addition, a recent study revealed an association between altered metabolism and the severity of chemotherapy-induced gastrointestinal and hematologic toxicity[8]. Although evidence have indicated that some metabolites

[1]Department of Anesthesia, The Sixth Affiliated Hospital, Sun Yat-sen University, Guangzhou, Guangdong, China. [2]Biomedical Innovation Center, The Sixth Affiliated Hospital, Sun Yat-sen University, Guangzhou, Guangdong, China. [3]Department of General Surgery (Colorectal Surgery), The Sixth Affiliated Hospital, Sun Yat-sen University, Guangzhou, Guangdong, China. [4]Guangdong Provincial Key Laboratory of Colorectal and Pelvic Floor Diseases, The Sixth Affiliated Hospital, Sun Yat-sen University, Guangzhou, Guangdong, China. [5]Key Laboratory of Human Microbiome and Chronic Diseases (Sun Yat-sen University), Ministry of Education, Guangzhou, Guangdong, China. [6]State Key Laboratory of Oncology in South China, Guangzhou, Guangdong, China. [7]These authors contributed equally: Hongyu Xie, Jingyi Yang, Jinjie Wu. ✉e-mail: jinsq@mail.sysu.edu.cn; lanping@mail.sysu.edu.cn; hezh5@mail.sysu.edu.cn

such as short-chain fatty acids are associated with chemotherapy-induced side effects, the role of tryptophan metabolism in chemotherapy-induced side effects is unclear.

Chemotherapy-induced side effects, especially intestinal injury, are also associated with alterations in the gut microbiota[9]. A previous study revealed an association between the microbiome and chemotherapy-induced gastrointestinal toxicity in children with acute lymphoblastic leukemia[10]. Enrichment of bacterial taxa *Lachnospiraceae* and *Enterococcaceae* was associated with postradiation restoration of gastrointestinal toxicity[11]. The supplementation of probiotics has also been shown to alleviate chemotherapy-induced toxicity[8,12,13]. However, the underlying mechanism through which the microbiota is modulated during chemotherapy-induced intestinal toxicity remains poorly understood.

Here, we show that specific tryptophan metabolites are associated with chemotherapy-induced side effects through targeted metabolomics analysis of CRC patient serum. Using distinct CRC mouse models, we demonstrate the contribution of specific metabolite to chemotherapy-induced intestinal toxicity and characterize the associated alterations in microbial components. Furthermore, we elucidate the underlying mechanism, suggesting that modulating tryptophan metabolic pathway offers a therapeutic strategy to improve tolerance to chemotherapy.

## Results

### Chemotherapy-induced intestinal toxicity is associated with the accumulation of L-kynurenine

To assess the role of tryptophan metabolism in chemotherapy-induced side effects, we established a clinical cohort and collected plasma from sixty-four patients who received oxaliplatin-based chemotherapy for targeted metabolomics of tryptophan metabolism (Fig. 1A). On the basis of the National Cancer Institute's Common Terminology Criteria for Adverse Events (NCI-CTCAE v5.0), these patients were classified into a high-toxicity group ($n = 30$) and a low-toxicity group ($n = 34$). Although similar baseline characteristics were found in these two groups, significantly higher levels of C-reactive protein (CRP) and lower level of white blood cells (WBCs) and neutrophils were detected in patients from higher toxicity group (Table S1). More importantly, a distinct pattern of tryptophan metabolites was observed between the high- and low-toxicity groups (Fig. 1B). Alteration of these tryptophan metabolites were closely correlated with different clinical parameters (Fig. 1C). Specifically, only four metabolites, L-kynurenine (L-Kyn), picolinic acid (PA), nicotinic acid (NAcid), and 3-hydroxyanthranilic acid (3-HAA), were significantly upregulated in plasma from the high-toxicity group (Fig. 1D–G). The increase of L-kynurenine in plasma from higher toxicity group was the most obvious metabolite in tryptophan metabolic pathway (Fig. 1B). A higher concentration of L-kynurenine in the plasma was associated with the exacerbation of toxicity, higher levels of CRP, and lower WBC and neutrophil counts (Fig. 1C). Collectively, these data indicated that increased levels of L-kynurenine were associated with chemotherapy-induced side effects.

To further validate the role of tryptophan metabolites in chemotherapy-induced side effects, we established a chemotherapy-induced toxicity model. C57BL/6J mice were treated with high-dose oxaliplatin (20 mg/kg) or PBS control once a week (Fig. 1H). Compared with control mice, mice treated with high-dose oxaliplatin exhibited significantly greater weight loss, worsened clinical scores, reduced spleen index and shortened colon length (Fig. 1I–L and Fig. S1A). Intestinal toxicity is among the important manifestations of chemotherapy-induced side effects. We found that mice exposed to high-dose oxaliplatin exhibited significantly greater separation between the crypt bases and muscularis mucosa (Fig. 1M), indicating the exacerbation of intestinal edema. Moreover, significantly decreased cellular proliferation and increased apoptosis of crypt cells

were observed in mice treated with high-dose oxaliplatin (Fig. 1N, O). Immunofluorescence analysis and transcript detection of intestinal tissues also revealed significant downregulation of MUC2 in mice with increased intestinal toxicity (Fig. 1P). To evaluate the alteration of tryptophan metabolites in a chemotherapy-induced toxicity mouse model, plasma from mice was collected to determine the targeted metabolome of tryptophan metabolism. Consistently, oxaliplatin-induced intestinal toxicity in mice was associated with a significant upregulation of L-kynurenine (Fig. 1Q). Taken together, these data indicated that oxaliplatin-induced intestinal toxicity was associated with alterations of tryptophan metabolites, especially the accumulation of L-kynurenine.

### Oxaliplatin-induced toxicity depends on IDO1 expression

To validate the role of L-kynurenine in oxaliplatin-induced toxicity, mice were gavaged with L-kynurenine three times a week, followed by oxaliplatin challenge, as shown in Fig. 2A. The toxicity induced by oxaliplatin, including severe weight loss, worsened clinical scores, decreased survival rate and decreased spleen index, was exacerbated in mice treated with L-kynurenine (Fig. 2B–E and Fig. S1B). In addition, mice treated with oxaliplatin and L-kynurenine exhibited obvious exacerbation of intestinal toxicity, such as shortened colon length, severe intestinal edema, decreased cellular proliferation, increased apoptosis of crypt cells and impairment of the intestinal barrier (Fig. 2F–J). Similar exacerbation of intestinal toxicity induced by oxaliplatin was also observed in a subcutaneous tumor xenograft model with L-kynurenine intervention (Fig. S1C–L). However, the chemotherapeutic potential of oxaliplatin was not affected (Fig. S1M, N). These data indicated that the accumulation of L-kynurenine aggravated oxaliplatin-induced intestinal toxicity in mice.

However, how L-kynurenine accumulates during chemotherapy remains unclear. A previous study demonstrated that L-kynurenine is converted from tryptophan via the key rate-limiting enzyme indoleamine 2,3-dioxygenase 1 (IDO1) (Fig. 2K). Significant upregulation of IDO1 expression was detected in mouse intestinal tissues (Fig. 2L), indicating the important role of IDO1 in chemotherapy-induced toxicity. Thus, we constructed a chemotherapy-induced toxicity model in *Ido1* knockout (*Ido1*[-/-]) mice. *Ido1*[-/-] mice and wild-type (WT) mice were injected with a high dose of oxaliplatin on days 0 and 7 (Fig. 2M). Strikingly, severe systemic toxicity in WT mice treated with oxaliplatin was significantly alleviated in *Ido1*[-/-] mice (Fig. 2N–P and Fig. S2A). Compared with those of WT mice, significant improvements in intestinal toxicity were also observed in *Ido1*[-/-] mice after treatment with a high dose of oxaliplatin (Fig. 2Q–U). Serum L-kynurenine levels were significantly lower in *Ido1*[-/-] mice than in control mice both before and after OXA treatment (Fig. S2B). Collectively, these data indicated that oxaliplatin-induced toxicity depended on the expression of IDO1. Targeted intervention of IDO1 may be an important therapeutic strategy to alleviate the intestinal toxicity induced by oxaliplatin.

### IFNγ-mediated upregulation of IDO1 exacerbates chemotherapy-induced toxicity

Given that previous studies have demonstrated that interferon-gamma (IFNγ) is a key upstream factor that induces IDO1 expression[14,15], we further investigated its role in oxaliplatin-induced toxicity. First, ELISA revealed significantly elevated levels of IFNγ in both serum and fecal samples of WT mice following treatment with high-dose oxaliplatin (Fig. 3A). To further identify the cellular sources of IFNγ expression, we performed flow cytometric analysis on various immune cell populations and intestinal epithelial cells isolated from oxaliplatin-treated mice. Our results demonstrated a significant increase in IFNγ production specifically by CD8[+] T cells (Fig. 3B). To elucidate the functional role of the IFNγ-IDO1 axis in chemotherapy toxicity, we administered a specific neutralizing antibody against IFNγ prior to high-dose oxaliplatin treatment (Fig. 3C). Compared with the control group without

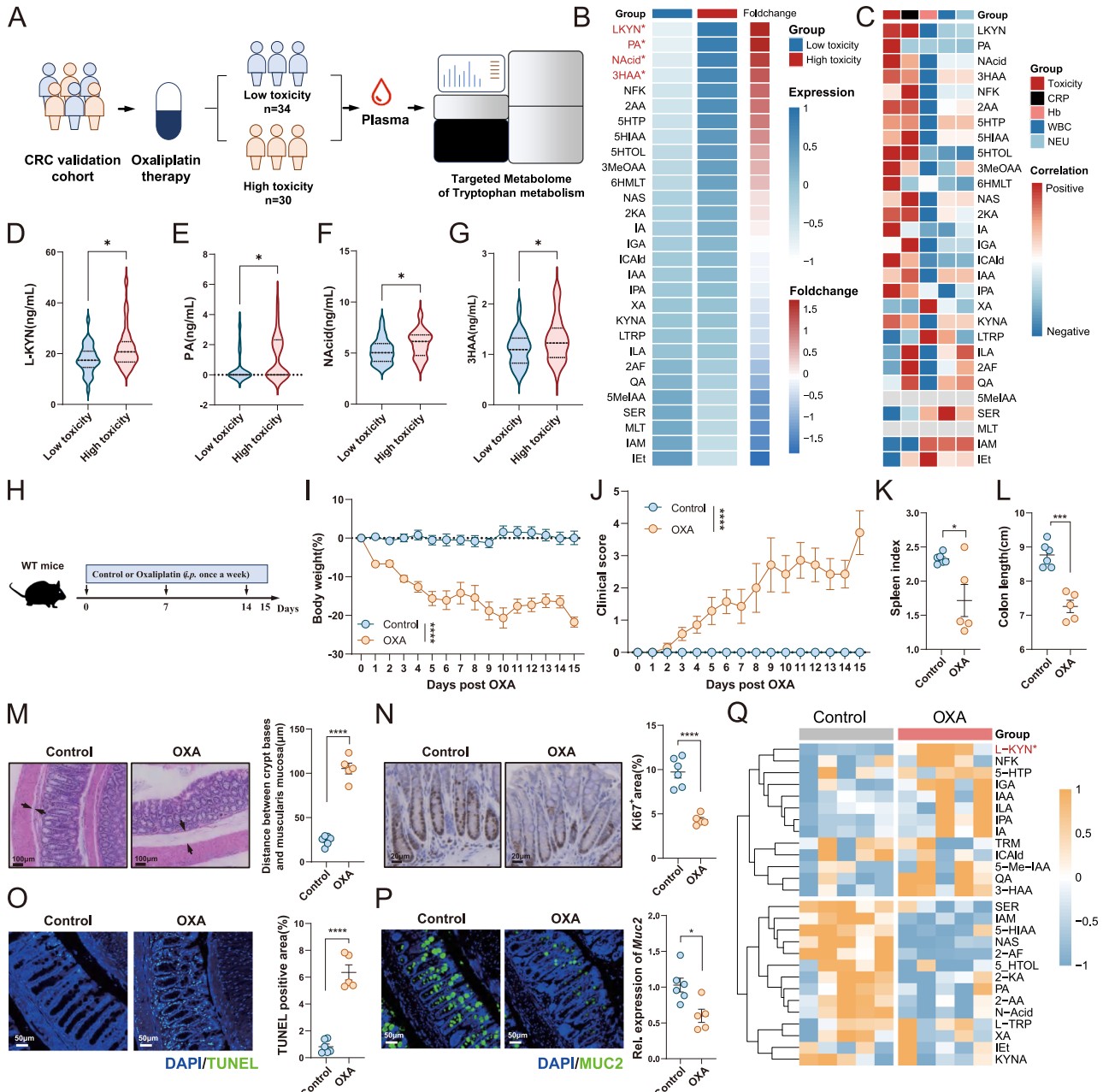

**Fig. 1 | Chemotherapy-induced toxicity is associated with higher level of L-kynurenine. A** Establishment of clinical cohort including 30 CRC patients with higher toxicity and 34 CRC patients with lower toxicity for targeted metabolome of tryptophan metabolism. **B** Heatmap of tryptophan-associated metabolites in patients with higher and lower toxicity. **C** Association of tryptophan-associated metabolites and clinical parameters. **D–G** Concentration of different significantly altered metabolites (L-kynurenine, L-KYN; Picolinic acid, PA; Nicotinic acid, NAcid; 3-hydroxyanthranilic acid, 3HAA). **H** Schematic diagram showing the process of oxaliplatin toxicity mice model treated with high-dose oxaliplatin or PBS control. Changes of body weight (**I**), clinical score (**J**), spleen index (**K**) and colon length (**L**). **M** Representative histopathological images of colon (left) and quantification for the gaps between crypt bases and muscularis mucosa (right). Arrows indicate gaps between crypt bases and muscularis mucosa. Scale bar = 100 μm.

**N** Immunohistochemical staining of Ki67 in intestinal tissue. Representative images (left) and the quantification of Ki67 area (right). Scale bar = 20 μm. **O** Immunofluorescence analysis of TUNEL in intestinal tissues. Representative images (left) and the quantification of TUNEL positive area (right). Scale bar = 50 μm. **P** Immunofluorescence analysis and transcript expression of MUC2 in intestinal tissues. Representative images (left) and the quantification of MUC2 expression (right). Scale bar = 50 μm. **I–P** $n = 6$ mice for control group and $n = 5$ mice for OXA group. **Q** Heatmap of tryptophan-associated metabolites in serum from mice treated with oxaliplatin and PBS control. ($n = 5$ for both group). Quantitative data are expressed as the mean ± standard error of the mean (S.E.M.). *$P < 0.05$, **$P < 0.01$, ***$P < 0.001$, ****$P < 0.0001$. $P$ values were determined by two-way ANOVA with Šídák's multiple comparisons test (**I**, **J**) and unpaired two-tailed Student's t-tests (**K–P**). Source data are provided as a Source Data file.

IFNγ neutralization, mice with IFNγ neutralization displayed significant improvements in systemic toxicity indicators (including weight loss, activity status) and intestinal toxicity parameters (including histopathological damage and colon length) (Fig. 3D–K and Fig. S2C). Critically, IFNγ neutralization significantly reduced IDO1 expression in intestinal tissues (Fig. S2D), directly confirming the IFNγ-dependent regulation of IDO1 in vivo. We also measured serum levels of L-kynurenine and found that IFNγ neutralization significantly reduced serum L-kynurenine levels compared with those in the control group (Fig. S2E). Collectively, these data indicate that the oxaliplatin triggers the production of IFNγ, which potently upregulates IDO1 expression, ultimately leading to severe chemotherapy-induced toxicity.

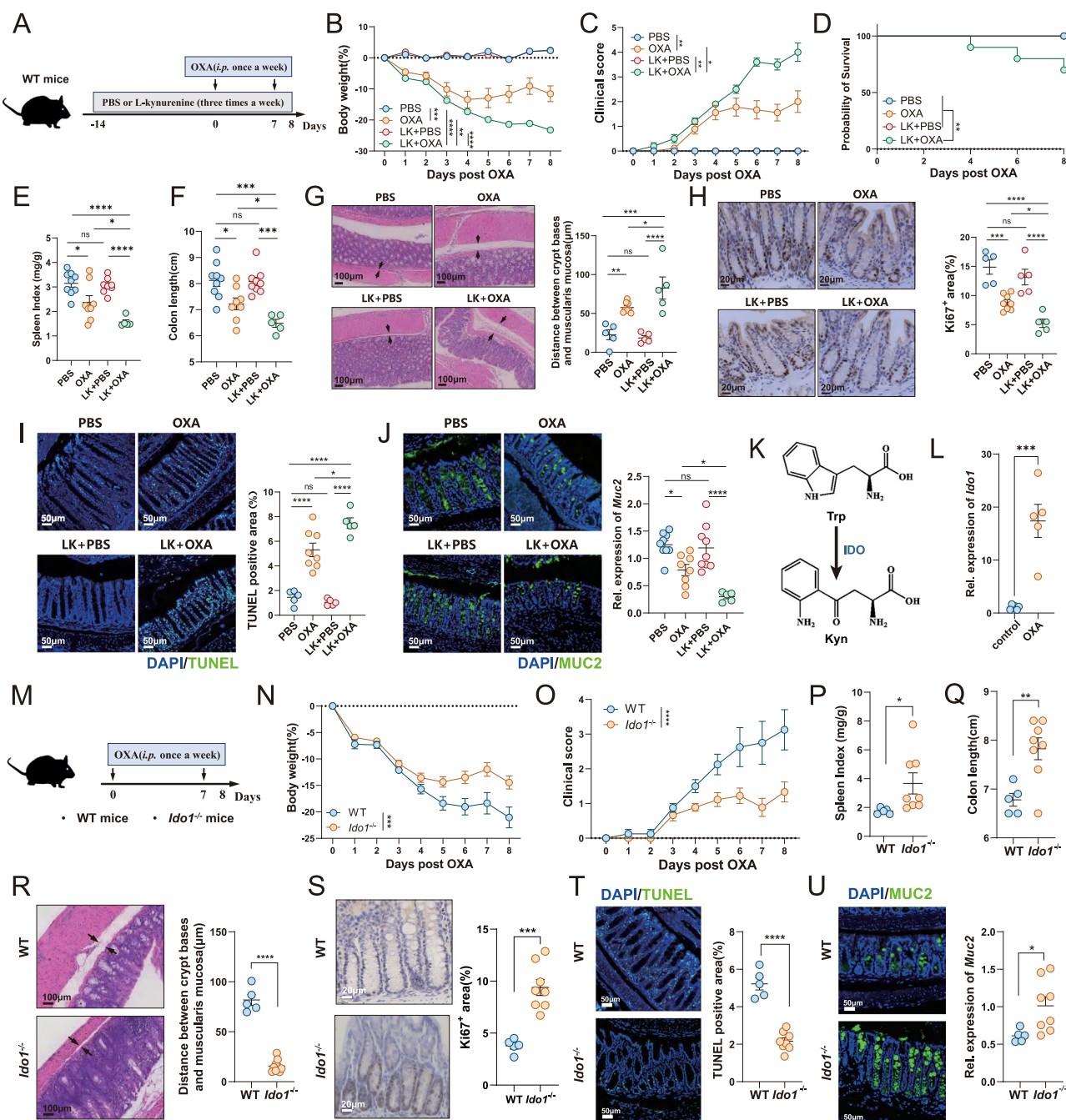

**Fig. 2 | Chemotherapy-induced toxicity depends on the expression of IDO1.**
**A** Schematic diagram showing the process of oxaliplatin toxicity mice model treated with L-kynurenine or PBS control. Changes of body weight (**B**), clinical score (**C**), survival rate (**D**), spleen index (**E**) and colon length (**F**). ($n = 9$ mice for PBS and LK + PBS group, $n = 8$ mice for OXA group, $n = 5$ mice for LK + OXA group). **G** Representative histopathological images of colon (left) and quantification for the gaps between crypt bases and muscularis mucosa (right). Arrows indicate gaps between crypt bases and muscularis mucosa. Scale bar = 100 μm.
**H** Immunohistochemical staining of Ki67 in intestinal tissue. Representative images (left) and the quantification of Ki67 area (right). Scale bar = 20 μm.
**I** Immunofluorescence analysis of TUNEL in intestinal tissues. Representative images (left) and the quantification of TUNEL positive area (right). Scale bar = 50 μm. **G–I** $n = 5$ mice for PBS, $n = 8$ mice for OXA group, $n = 5$ mice for LK + PBS group, $n = 5$ mice for LK + OXA group. **J** Immunofluorescence analysis and transcript expression of MUC2 in intestinal tissues. Representative images (left) and the quantification of MUC2 expression (right). Scale bar = 50 μm. ($n = 9$ mice for PBS, $n = 8$ mice for OXA group, $n = 9$ mice for LK + PBS group, $n = 5$ mice for LK + OXA group). **K** Schematic diagram of L-kynurenine metabolism from Tryptophan via IDO1. **L** Transcript expression of Ido1 in intestinal tissues from mice treated with

oxaliplatin and PBS control. $n = 6$ for control group, $n = 5$ for OXA group.
**M** Schematic diagram showing the process of *Ido1*[-/-] mice and WT mice treated with oxaliplatin. Changes of body weight (**N**), clinical score (**O**), spleen index (**P**) and colon length (**Q**). **R** Representative histopathological images of colon (left) and quantification for the gaps between crypt bases and muscularis mucosa (right). Arrows indicate gaps between crypt bases and muscularis mucosa. Scale bar = 100 μm. **S** Immunohistochemical staining of Ki67 in intestinal tissue. Representative images (left) and the quantification of Ki67 area (right). Scale bar = 20 μm. **T** Immunofluorescence analysis of TUNEL in intestinal tissues. Representative images (left) and the quantification of TUNEL positive area (right). Scale bar = 50 μm. **U** Immunofluorescence analysis and transcript expression of MUC2 in intestinal tissues. Representative images (left) and the quantification of MUC2 expression (right). Scale bar = 50 μm. **N–U** $n = 5$ mice for WT group, $n = 8$ for *Ido1*[-/-] group. Quantitative data are expressed as the mean ± standard error of the mean (S.E.M). *$P < 0.05$, **$P < 0.01$, ***$P < 0.001$, ****$P < 0.0001$; ns, not significant. $P$ values were determined by two-way ANOVA with Tukey's post hoc test (**B**, **C**), one-way ANOVA with Tukey's post hoc test (**D–J**), two-way ANOVA with Šídák's multiple comparisons test (**N**, **O**) and unpaired two-tailed Student's t-tests (**L**, **N–U**). Source data are provided as a Source Data file.

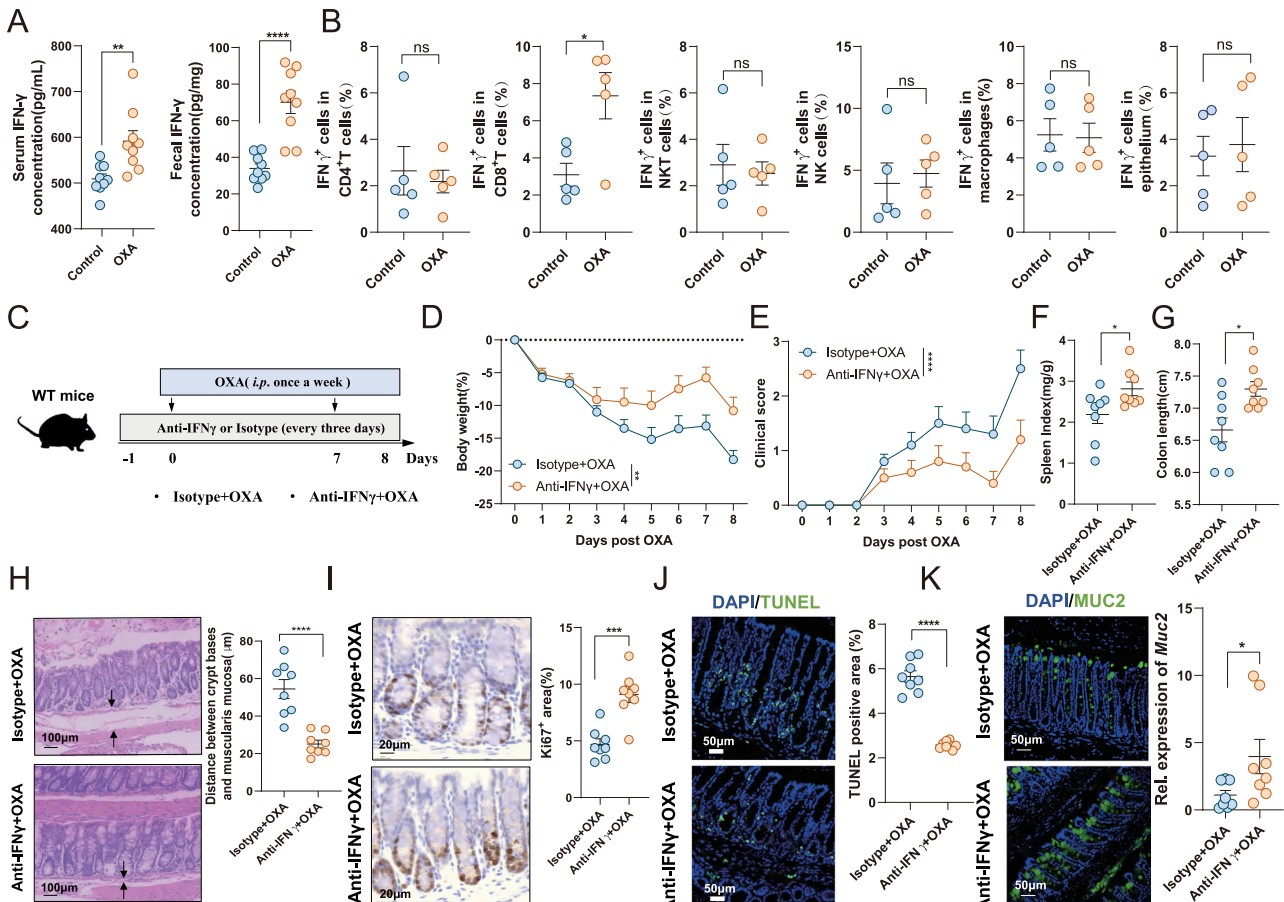

**Fig. 3 | IFNγ drives chemotherapy-induced toxicity via IDO1 upregulation.**
**A** Serum (left) and fecal (right) levels of IFNγ in control and OXA-treated mice. *n* = 9 mice per group. **B** Statistical analysis of the percentage of IFNγ-positive CD4⁺T, CD8⁺ T, NKT, NK, macrophage, epithelial cells by flow cytometry. *n* = 5 mice per group. **C** Schematic diagram showing the process of oxaliplatin toxicity mice model treated with Anti-IFNγ or Isotype control. Change of body weight (**D**), clinical score (**E**), spleen index (**F**) and colon length (**G**). **H** Representative histopathological images of colon (left) and quantification for the gaps between crypt bases and muscularis mucosa (right). Arrows indicate gaps between crypt bases and muscularis mucosa. Scale bar = 100 μm. **I** Immunohistochemical staining of Ki67 in intestinal tissue. Representative images (left) and the quantification of Ki67

area (right). **J** Immunofluorescence analysis of TUNEL in intestinal tissues. Representative images (left) and the quantification of TUNEL positive area (right). Scale bar = 50 μm. **K** Immunofluorescence analysis and transcript expression of MUC2 in intestinal tissues. Representative images (left) and the quantification of MUC2 expression (right). Scale bar = 50 μm. **D**–**K** *n* = 8 mice per group. Quantitative data are expressed as the mean ± standard error of the mean (S.E.M). *$P < 0.05$, **$P < 0.01$, ***$P < 0.001$, ****$P < 0.0001$; ns, not significant. *P* values were determined by two-way ANOVA with Šídák's multiple comparisons test (**D**, **E**) and unpaired two-tailed Student's t-tests (**A**, **B**, **F**–**K**). Source data are provided as a Source Data file.

## Alteration of IDO1 reshapes intestinal cellular components

To elucidate the role of IDO1 in intestinal tissues, we performed scRNA-seq on intestinal tissues from *Ido1*⁻/⁻ and *Ido1*⁻/⁺ mice. Analysis of combined intestinal tissues from *Ido1*⁻/⁻ and *Ido1*⁻/⁺ mice revealed 13 different cell clusters on the basis of the expression levels of the most variable genes (Fig. 4A). The cell clusters identified in intestinal tissues primarily consisted of myeloid cells (*Ly6g*⁺, *Ly6a*⁺ and *Tgm3*⁺), enterocytes (enterocyte 1: *Slc13a1*⁺, *Slc10a2*⁺ and *Prap1*⁺; enterocyte 2: *Krt36*⁺, *Sele*⁺ and *Krt16*⁺; enterocyte 3: *Gsdmc4*⁺, *Krt84*⁺ and *Krtap6-5*⁺), intestinal goblet cells (intestinal goblet cells 1: *Muc5ac*⁺, *Muc2*⁺ and *Slc12a8*⁺; intestinal goblet cells 2: *Clca1*⁺, *Fcgbp*⁺ and *Tff3*⁺), B lymphocytes (*Cd74*⁺, *Cd79a*⁺ and *Cd79b*⁺), T lymphocytes (*Crtam*⁺, *Klrc1*⁺, *Trgc4*⁺ and *Pdcd1*⁺), fibroblasts (*Prrx1*⁺, *Clec3b*⁺, *Dcn*⁺ and *Ogn*⁺), enteroendocrine cells (*Chga*⁺, *Chgb*⁺, *Pyy*⁺ and *Sst*⁺) and lymphatic endothelial cells (*Prox1*⁺, *Kdr*⁺ and *Myct1*⁺) (Fig. 4B). Specifically, the most prominent changes observed in *Ido1*⁻/⁻ mice included a significant expansion of myeloid cells and a significant reduction in enterocyte 3 (Fig. 4C). Several genes associated with inflammatory and chemotactic processes were clearly downregulated in myeloid cells from *Ido1*⁻/⁻ mice (Fig. 4D). Functional enrichment analysis revealed that alterations in these myeloid cells were associated with several inflammatory

pathways, such as the PI3K-Akt signaling pathway, cytokine–cytokine receptor interactions, the chemokine signaling pathway, complement and coagulation cascades, and the IL-17 signaling and TNF signaling pathways (Fig. 4E). To further evaluate the cellular alteration upon changes in L-kynurenine metabolism, cell–cell interactions were subsequently constructed. Interestingly, myeloid cells played important roles in cellular interactions in the intestinal microenvironment (Fig. 4F). Strong interactions were observed between myeloid cells and enterocytes (Fig. 4F). Thus, we also characterized the alteration of enterocytes due to the role of L-kynurenine in intestinal toxicity. Consistent with the mouse phenotype, several genes associated with inflammatory processes, chemotactic processes and apoptosis were downregulated in *Ido1*⁻/⁻ mice (Fig. 4G). Alteration of these three types enterocytes was associated with focal adhesion, cell adhesion molecules and ECM-receptor interaction (Fig. 4H). In addition, some inflammatory pathways such as *Staphylococcus aureus* infection, complement and coagulation cascades and intestinal immune network for IgA production were involved in the intestinal microenvironment (Fig. 4H). Collectively, these data emphasize the crucial role of myeloid cells and enterocytes in the L-kynurenine-associated chemotherapy toxicity.

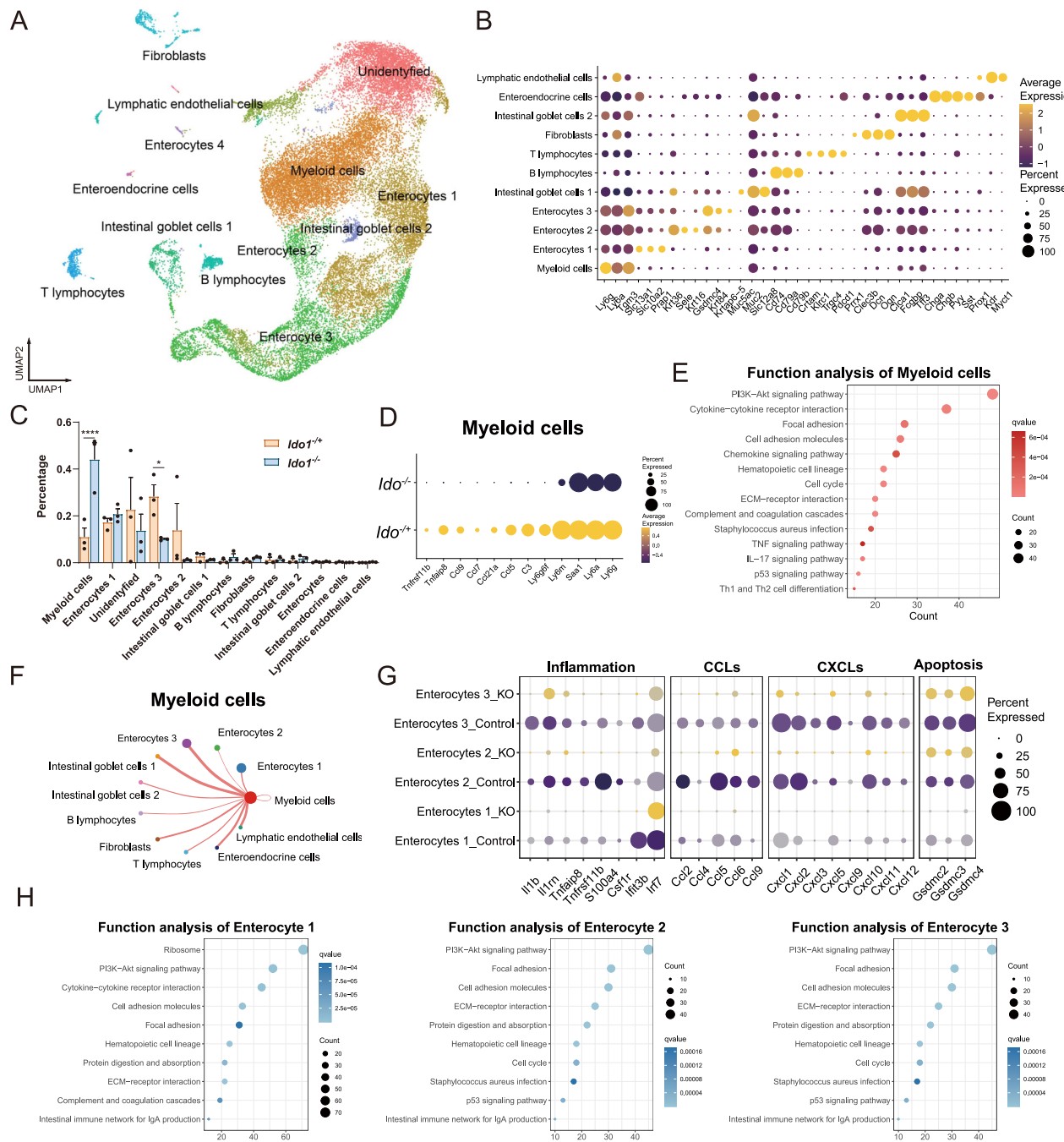

**Fig. 4 | Alteration of L-kynurenine metabolism upregulate the intestinal infiltration of myeloid cells. A** UMAP of intestinal single cells collected from *Ido1⁻/⁻* mice and *Ido1⁻/⁺* mice. *n* = 3 mice per group. **B** The bubble plot showing cluster-specific gene expression in different cell clusters. **C** Proportion of different cell clusters in intestinal tissues from *Ido1⁻/⁻* mice and *Ido1⁻/⁺* mice. **D** The bubble plot showing myeloid cells-specific gene expression in intestinal tissues from *Ido1⁻/⁻* mice and *Ido1⁻/⁺* mice. **E** Functional analysis of the differential genes in myeloid cells from *Ido1⁻/⁻* mice and *Ido1⁻/⁺* mice. **F** Cellular interaction between different immune cell types and myeloid cells. **G** The bubble plot showing the specific genes expression in different subtypes of enterocytes in the intestinal tissues from *Ido1⁻/⁻* mice and *Ido1⁻/⁺* mice. **H** Functional analysis of the differential genes in different subtypes of enterocytes from *Ido1⁻/⁻* mice and *Ido1⁻/⁺* mice. Quantitative data are expressed as the mean ± standard error of the mean (S.E.M). *$P$ < 0.05, ****$P$ < 0.0001. $P$ values were determined by two-way ANOVA with Šídák's multiple comparisons test (**C**). Source data are provided as a Source Data file.

## L-kynurenine metabolism in myeloid cells is responsible for chemotherapy-induced toxicity

Although L-kynurenine-induced chemotherapy toxicity is associated with alterations in myeloid cells and enterocytes, the role of these cellular components in L-kynurenine biosynthesis during chemotherapy-induced toxicity remains unclear. To confirm the role of myeloid cell-associated IDO1 in chemotherapy toxicity, we constructed *Ido1^flox/flox Lyz2-Cre* mice, which exhibited specific depletion of IDO1 in myeloid cells. A chemotherapy-induced toxicity model was established with *Ido1^flox/flox Lyz2-Cre* and *Ido1^flox/flox* mice, which were treated with oxaliplatin once a week (Fig. 5A). Compared with *Ido1^flox/flox* mice, *Ido1^flox/flox Lyz2-Cre* mice treated with oxaliplatin exhibited improvements in weight loss, clinical score and spleen index (Fig. 5B–D). Histopathological analysis also revealed a significant alleviation of intestinal toxicity in *Ido1^flox/flox Lyz2-Cre* mice (Fig. 5E–H). To evaluate the role of

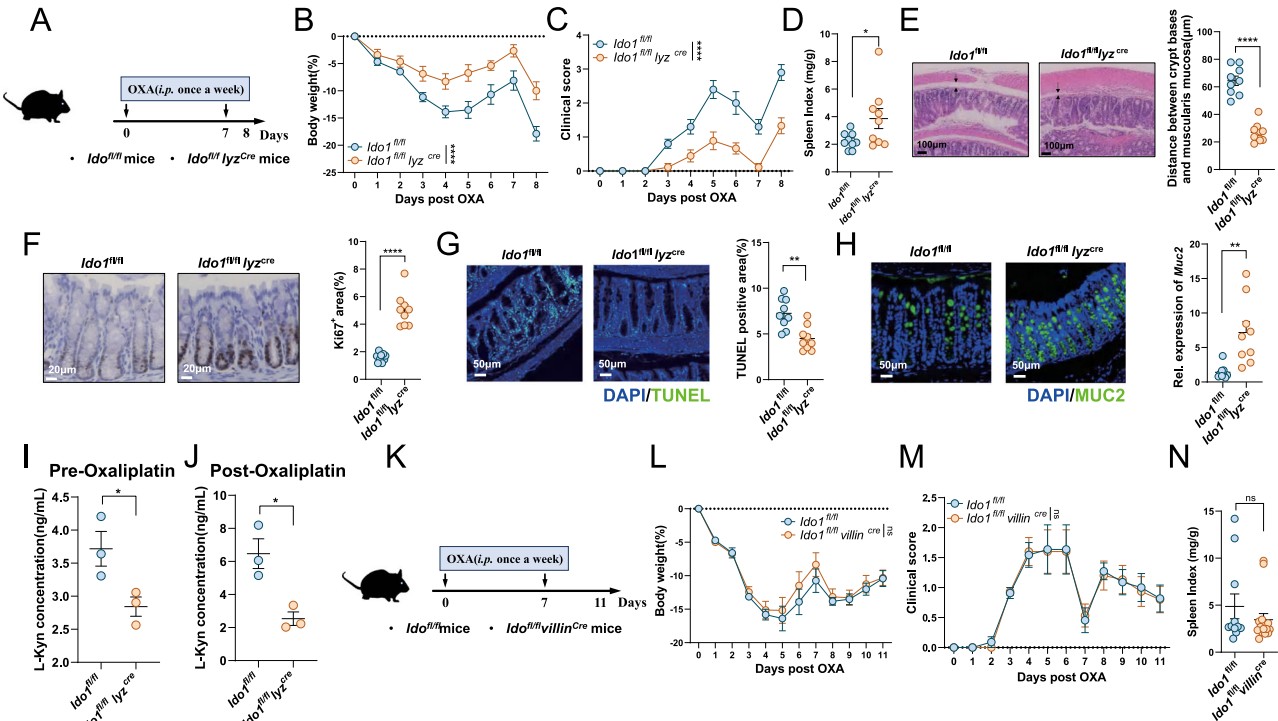

**Fig. 5 | Downregulation of L-kynurenine in myeloid cells improved the chemotherapy-induced toxicity. A** Schematic diagram showing the process of oxaliplatin toxicity mice model with *Ido1*[flox/flox] *Lyz2-Cre* mice and *Ido1*[flox/flox] mice. Changes of body weight (**B**), clinical score (**C**) and spleen index (**D**). **E** Representative histopathological images of colon (left) and quantification for the gaps between crypt bases and muscularis mucosa (right). Arrows indicate gaps between crypt bases and muscularis mucosa. Scale bar = 100 μm. **F** Immunohistochemical staining of Ki67 in intestinal tissue. Representative images (left) and the quantification of Ki67 area (right). Scale bar = 20 μm. **G** Immunofluorescence analysis of TUNEL in intestinal tissues. Representative images (left) and the quantification of TUNEL positive area (right). Scale bar = 50 μm. **H** Immunofluorescence analysis and transcript expression of MUC2 in intestinal tissues. Representative images (left) and the quantification of MUC2 expression (right). Scale bar = 50 μm. **B**–**H** *n* = 9 mice

for both groups. **I** Serum L-kynurenine concentration in *Ido1*[flox/flox] *Lyz2-Cre* mice and *Ido1*[flox/flox] mice prior to oxaliplatin treatment. (*n* = 3 for both groups). **J** Serum L-kynurenine concentration in *Ido1*[flox/flox] *Lyz2-Cre* mice and *Ido1*[flox/flox] mice after oxaliplatin treatment. (*n* = 3 for both groups). **K** Schematic diagram showing the process of oxaliplatin toxicity mice model with *Ido1*[flox/flox] *Villin-Cre* mice and *Ido1*[flox/flox] mice. Changes of body weight (**L**), clinical score (**M**) and spleen index (**N**). *n* = 11 for *Ido1*[flox/flox] group, *n* = 15 for *Ido1*[flox/flox] *Villin-Cre* group. Quantitative data are expressed as the mean ± standard error of the mean (S.E.M.). *$P < 0.05$, **$P < 0.01$, ***$P < 0.001$, ****$P < 0.0001$; ns, not significant. *P* values were determined by two-way ANOVA with Šídák's multiple comparisons test (**B**, **C**, **L**, **M**) and unpaired two-tailed Student's t-tests (**D**–**J**, **N**). Source data are provided as a Source Data file.

myeloid cell-derived IDO1 in tumor growth, a subcutaneous tumor xenograft model was subsequently established. Similar alleviation of toxicity was observed in *Ido1*[flox/flox] *Lyz2-Cre* mice (Fig. S3A–J). We further measured serum levels of L-kynurenine in both *Ido1*[flox/flox] *Lyz2-Cre* and *Ido1*[flox/flox] mice, with or without OXA treatment. Consistent with the specific deletion of IDO1 in myeloid cells, serum L-kynurenine levels were significantly lower in *Ido1*[flox/flox] *Lyz2-Cre* mice than in control mice (Fig. 5I, J). To explore whether the expression of IDO1 in enterocytes modulates chemotherapy toxicity, we established a mouse model by using *Ido1*[flox/flox] *Villin-Cre* mice, which exhibited specific depletion of IDO1 in the epithelium (Fig. 5K). Similar trends in weight loss, clinical score and spleen index were observed in *Ido1*[flox/flox] *Villin-Cre* and *Ido1*[flox/flox] mice (Fig. 5L–N). These data confirmed that downregulation of IDO1 in myeloid cells but not in the epithelium improved chemotherapy-induced toxicity.

## L-kynurenine metabolism modulates oxaliplatin-induced intestinal toxicity via the gut microbiota

Although alterations in L-kynurenine metabolism are associated with chemotherapy-induced toxicity, the underlying mechanism through which L-kynurenine metabolism modulates oxaliplatin-induced intestinal toxicity remains unclear. Previous studies have demonstrated that the gut microbiota plays a critical role in oxaliplatin-

induced toxicity[8]. Thus, we established a chemotherapy-induced toxicity model with *Ido1*[-/-] mice and WT mice, both of which were treated with an antibiotic cocktail (Fig. 6A). The improvements in weight loss, clinical scores, spleen index and colon length in *Ido1*[-/-] mice disappeared after treatment with antibiotics, which was similar to the phenotypes in WT mice (Fig. 6B–E and Fig. S4A). Moreover, alleviation of intestinal toxicity induced by oxaliplatin was exclusively observed in *Ido1*[-/-] mice treated with the PBS control (Fig. 6F–I). Exacerbation of intestinal edema, decreased cellular proliferation, increased apoptosis of crypt cells and impairment of the intestinal barrier were detected in *Ido1*[-/-] mice treated with antibiotics and in WT mice (Fig. 6F–I). These data demonstrated that the influence of L-kynurenine metabolism on oxaliplatin-induced intestinal toxicity was associated with alterations of gut microbiota.

To further validate the role of the microbiota in oxaliplatin-induced toxicity, fecal microbiota transplantation (FMT) experiments were subsequently performed. Fecal samples collected from *Ido1*[-/-] mice or WT mice were transferred to recipient WT mice three times a week after 7 days of antibiotic treatment (Fig. 6J). The mice were challenged with a high dose of oxaliplatin once a week after FMT intervention (Fig. 6J). As expected, the recipient mice receiving microbiota from *Ido1*[-/-] mice exhibited significantly alleviated weight loss, clinical scores, spleen index and colon length (Fig. 6K–N and Fig. S4B). Moreover, a significant improvement in oxaliplatin-induced

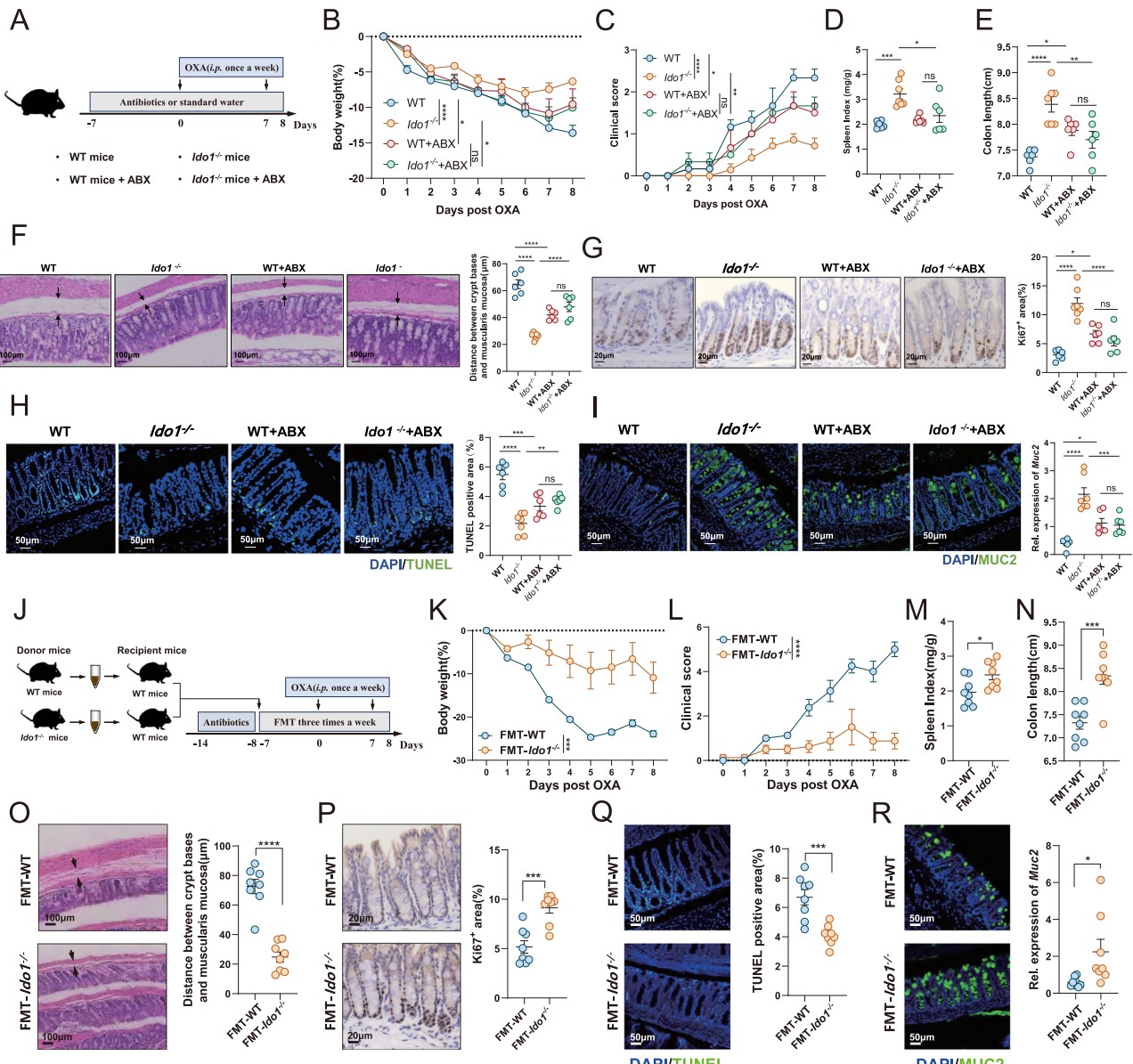

**Fig. 6 | L-kynurenine metabolism modulate the chemotherapy-induced toxicity via gut microbiota. A** Schematic diagram showing the process of oxaliplatin toxicity mice model treated with antibiotics or PBS control. Changes of body weight (**B**), clinical score (**C**), spleen index (**D**) and colon length (**E**). **F** Representative histopathological images of colon (left) and quantification for the gaps between crypt bases and muscularis mucosa (right). Arrows indicate gaps between crypt bases and muscularis mucosa. Scale bar = 100 μm. **G** Immunohistochemical staining of Ki67 in intestinal tissue. Representative images (left) and the quantification of Ki67 area (right). Scale bar = 20 μm. **H** Immunofluorescence analysis of TUNEL in intestinal tissues. Representative images (left) and the quantification of TUNEL positive area (right). Scale bar = 50 μm. **I** Immunofluorescence analysis and transcript expression of MUC2 in intestinal tissues. Representative images (left) and the quantification of MUC2 expression (right). Scale bar = 50 μm. **B**–**I** *n* = 6 mice for WT group, *n* = 7 mice for *Ido1*⁻ group, *n* = 6 mice for WT + ABX group, and *n* = 6 mice for *Ido1*⁻ + ABX group. **J** Schematic diagram showing the process of oxaliplatin toxicity mice model treated with feces from *Ido1*⁻ mice and WT mice. Changes of body weight (**K**), clinical

score (**L**), spleen index (**M**) and colon length (**N**). **O** Representative histopathological images of colon (left) and quantification for the gaps between crypt bases and muscularis mucosa (right). Arrows indicate gaps between crypt bases and muscularis mucosa. Scale bar = 100 μm. **P** Immunohistochemical staining of Ki67 in intestinal tissue. Representative images (left) and the quantification of Ki67 area (right). Scale bar = 20 μm. **Q** Immunofluorescence analysis of TUNEL in intestinal tissues. Representative images (left) and the quantification of TUNEL positive area (right). Scale bar = 50 μm. **R** Immunofluorescence analysis and transcript expression of MUC2 in intestinal tissues. Representative images (left) and the quantification of MUC2 expression (right). Scale bar = 50 μm. **K**–**R** *n* = 8 mice for FMT-WT and FMT- *Ido1*⁻ group. Quantitative data are expressed as the mean ± standard error of the mean (S.E.M). *$P < 0.05$, **$P < 0.01$, ***$P < 0.001$, ****$P < 0.0001$; ns, not significant. *P* values were determined by two-way ANOVA with Tukey's post hoc test (**B**, **C**), one-way ANOVA with Tukey's post hoc test (**D**–**I**), two-way ANOVA with Šídák's multiple comparisons test (**K**, **L**) and unpaired two-tailed Student's t-tests (**M**–**R**). Source data are provided as a Source Data file.

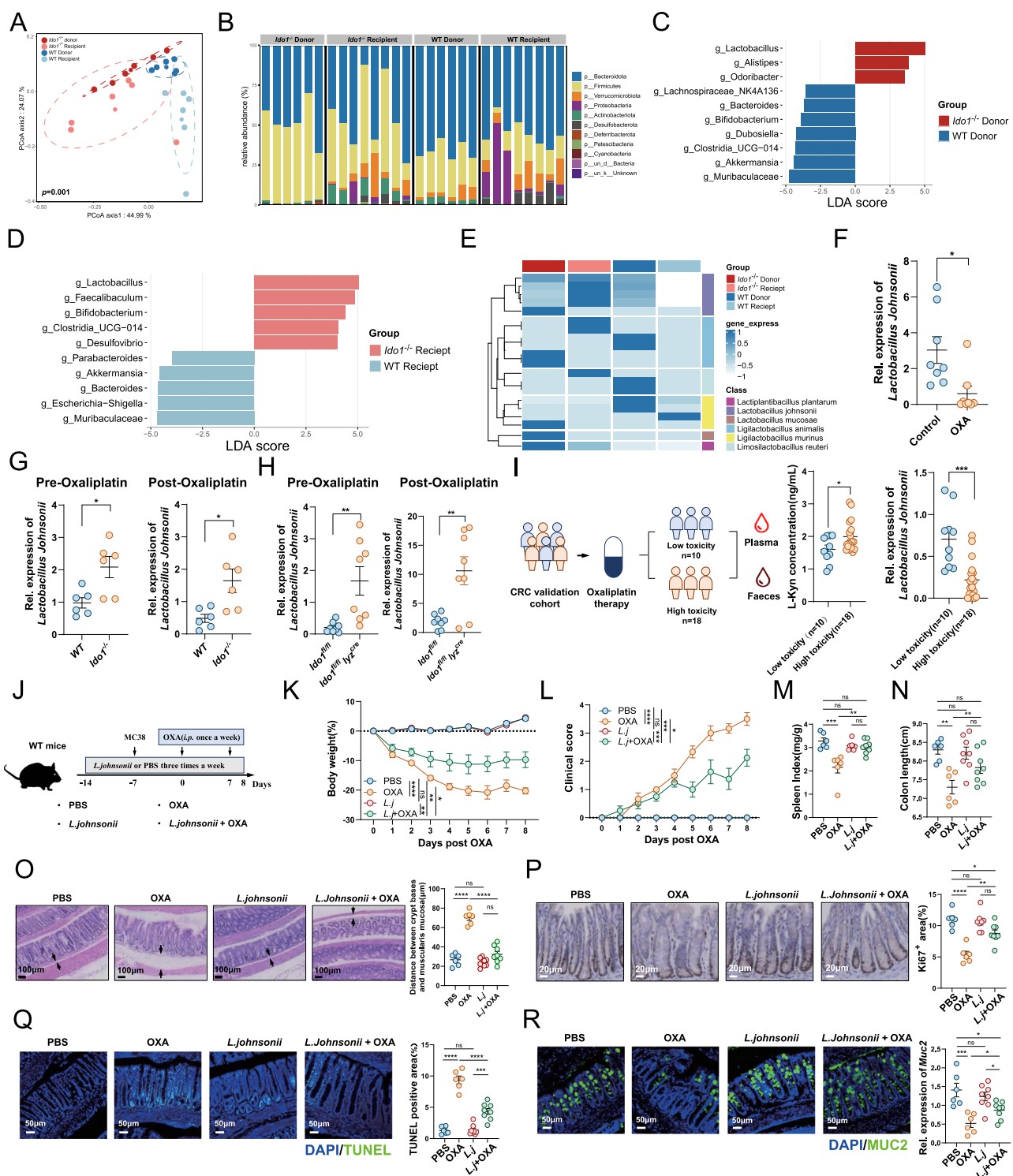

intestinal toxicity was also observed in recipient mice treated with microbiota from *Ido1$^{-/-}$* mice (Fig. 6O–R). Therefore, these data indicated that the metabolism of L-kynurenine modulated oxaliplatin-induced intestinal toxicity via alteration of the gut microbiota.

### Downregulation of L-kynurenine promotes the colonization of *L. johnsonii* to alleviate oxaliplatin toxicity

To clarify how L-kynurenine metabolism modulates the gut microbiota, fecal samples from donor and recipient mice were collected for 16S ribosomal RNA sequencing. Taxonomic analysis of the microbiome using principal coordinate analysis (PCoA) revealed that

different genotypes of mice exhibited significant clustering and separation of the microbiome (Fig. 7A). The microbial structures of donor and recipient mice with the same genotype were similar (Fig. 7A, B). An increased proportion of *Firmicutes* was detected in the feces of *Ido1$^{-/-}$* donor and recipient mice, whereas a greater proportion of *Bacteroidetes* was detected in WT mice (Fig. 7B). We subsequently detected marked alterations in microbes between different genotypes of donor mice and recipient mice. Specifically, significant accumulation of *Lactobacillus* was observed in *Ido1$^{-/-}$* donor and recipient mice (Fig. 7C, D). Further identification of microbial units by the Basic Local Alignment Search Tool (BLAST) revealed that a stronger signal of

**Fig. 7 | Accumulation of *L. johnsonii* in gut improved the chemotherapy-induced toxicity. A** Principal component analysis of microbial structure in feces from donor and recipient mice. **B** Relative abundance of microbiota at phylum level in feces from donor and recipient mice. **C** Alteration of genus in feces from *Ido1*[-/-] and WT donor mice. **D** Alteration of genus in feces from *Ido1*[-/-] and WT recipient mice. **E** Heatmap of *Lactobacillus* species in feces from donor and recipient mice. **F** The relative abundance of *L. johnsonii* in the feces of WT mice. **G** The relative abundance of *L. johnsonii* in the feces of *Ido1*[-/-] and WT mice before (left) and after (right) oxaliplatin treatment. (*n* = 6 for both groups). **H** The relative abundance of *L. johnsonii* in the feces of *Ido1*[flox/flox] *Lyz2-Cre* and *Ido1*[flox/flox] mice before (left) and after (right) oxaliplatin treatment. (*n* = 8 for both groups). **I** Establishment of a clinical cohort (10 low-toxicity vs 18 high-toxicity CRC patients) for assessing serum L-kynurenine and fecal *L. johnsonii* abundance. **J** Schematic diagram showing the process of subcutaneous tumor xenograft model treated with *L. johnsonii* or PBS control. Changes of body weight (**K**), clinical score (**L**), spleen index (**M**) and colon length (**N**). **O** Representative histopathological images of colon (left) and

quantification for the gaps between crypt bases and muscularis mucosa (right). Arrows indicate gaps between crypt bases and muscularis mucosa. Scale bar = 100 μm. **P** Immunohistochemical staining of Ki67 in intestinal tissue. Representative images (left) and the quantification of Ki67 area (right). Scale bar = 20 μm. **Q** Immunofluorescence analysis of TUNEL in intestinal tissues. Representative images (left) and the quantification of TUNEL positive area (right). Scale bar = 50 μm. **R** Immunofluorescence analysis and transcript expression of MUC2 in intestinal tissues. Representative images (left) and the quantification of MUC2 expression (right). Scale bar = 50 μm. **K–R** *n* = 6 mice per PBS and OXA group, *n* = 8 mice per *L. johnsonii* and *L. johnsonii* + OXA group. Quantitative data are expressed as the mean ± standard error of the mean (S.E.M). *\*P* < 0.05, *\*\*P* < 0.01, *\*\*\*P* < 0.001, *\*\*\*\*P* < 0.0001; ns, not significant. *P* values were determined by the unpaired two-tailed Student's t-tests (**F–I**), two-way ANOVA with Tukey's post hoc test (**K**, **L**) and one-way ANOVA with Tukey's post hoc test (**M–R**). Source data are provided as a Source Data file.

*Lactobacillus johnsonii* (*L. johnsonii*) was detected in *Ido1*[-/-] donor and recipient mice (Fig. 7E). We subsequently evaluated the abundance of *L. johnsonii* in mouse feces using qPCR. As expected, oxaliplatin treatment significantly reduced the relative abundance of *L. johnsonii* in the feces of WT mice (Fig. 7F). In contrast, *Ido1*[-/-] and *Ido1*[flox/flox] *Lyz2-Cre* mice presented a higher baseline abundance of *L. johnsonii*, which was maintained after oxaliplatin treatment, resulting in consistently higher levels than those found in control group mice regardless of treatment (Fig. 7G, H). Moreover, in a clinical validation cohort of patients receiving oxaliplatin-based chemotherapy, we observed that serum L-kynurenine levels were significantly higher and that the abundance of fecal *L. johnsonii* was significantly lower in the high-toxicity group than those in the low-toxicity group (Fig. 7I). These results suggested that the downregulation of L-kynurenine metabolism enhanced the colonization of *L. johnsonii*.

To validate the role of *L. johnsonii* in oxaliplatin toxicity, we established a subcutaneous tumor xenograft model that was gavage with *L. johnsonii* three times a week followed by oxaliplatin treatment once a week (Fig. 7J). Notably, tumor size and tumor weight did not show significant differences between the groups, suggesting that supplementation with *L. johnsonii* did not affect the efficacy of oxaliplatin treatment (Fig. S5A–C). However, treatment with *L. johnsonii* significantly improved the oxaliplatin-induced toxicity, including the weight loss, clinical score, spleen index and colon length (Fig. 7K–N and Fig. S5D). Histopathological analysis also revealed a significant improvement of oxaliplatin-induced intestinal toxicity in mice treated with *L. johnsonii* (Fig. 7O–R). These data supported that downregulation of L-kynurenine metabolism promoted accumulation of *L. johnsonii*, therefore resulting in an alleviation of chemotherapy toxicity.

To investigate the effect of L-kynurenine on *L. johnsonii*, we first cultured the bacterium in vitro using medium supplemented with L-kynurenine. The results demonstrated that L-kynurenine significantly inhibited the growth of *L. johnsonii* (Fig. S5E). We subsequently performed transcriptome sequencing on L-kynurenine-treated and untreated *L. johnsonii*. Principal component analysis (PCA) of the sequencing data revealed distinct separations among the control group, the low-concentration L-kynurenine group (LKL), and the high-concentration L-kynurenine group (LKH) (Fig. S5F). In total, 1477 genes were annotated, and 32 genes were significantly downregulated in the LKL group compared with the control group (Fig. S5G). In the LKH group, 175 genes were significantly downregulated, whereas 82 genes were significantly upregulated compared with those in the control group (Fig. S5H). The functional annotation of the differentially expressed genes (DEGs) revealed that the DEGs were enriched primarily in processes related to adhesion, biofilm formation, exotoxin production, immune modulation, invasion mechanisms, motility, nutrient and metabolic factors, and stress response (Fig. S5I). After

treatment with L-kynurenine, we observed a significant downregulation of several genes essential for the survival and proliferation of *L. johnsonii*. Specifically, the expression of *AcpP*[16,17] (fatty acid synthesis), *LepB*[18,19] and *LspA*[20] (signal peptidases), *DltB*[21] (cell wall modification), and *RpoD*[22] (RNA polymerase) were decreased (Fig. S5J). These findings suggest that L-kynurenine may inhibit the growth of *L. johnsonii*, compromising its ability to colonize the gut.

## L-kynurenine exacerbates apoptosis via TNFα/JNK pathway activation and *L. johnsonii* intervention attenuates this effect

Studies have demonstrated that radiotherapy-induced intestinal damage is mediated by activation of the TNFα (tumor necrosis factor-α) /JNK (c-Jun N-terminal kinase) signaling pathway[23]. To elucidate the downstream mechanism through which L-kynurenine exacerbates chemotherapy-induced intestinal injury, we focused on the TNFα/JNK cascade—a key pathway regulating epithelial apoptosis. TNFα levels were markedly elevated in the intestinal tissue of L-kynurenine -treated mice (Fig. S6A). Importantly, administration of *L. johnsonii* reversed this effect, resulting in a marked downregulation of TNFα expression (Fig. S6A). To further clarify the underlying mechanism, we performed in vitro experiments using epithelial cells treated with L-kynurenine in combination with OXA. This combined treatment led to a significant increase in phosphorylated JNK (p-JNK) levels (Fig. S6B). Probiotic treatment with *L. johnsonii* effectively attenuated activation of p-JNK (Fig. S6B). Meanwhile, L-kynurenine/OXA exposure markedly increased epithelial apoptosis (Fig. S6C), and *L. johnsonii* significantly reduced epithelial apoptosis (Fig. S6D). Together, these findings indicate that *L. johnsonii* alleviates chemotherapy-induced intestinal damage by suppression of TNFα/JNK signaling, thus inhibiting epithelial apoptosis.

## Pharmacological and microbial therapeutic strategies attenuate IDO1/L-kynurenine-mediated chemotoxicity

On the basis of the protective effects observed in the genetic model, we further sought to confirm the therapeutic potential of pharmacologically inhibiting IDO1 using the specific inhibitor Epacadostat. WT mice were gavaged with PBS or Epacadostat twice a day and high-dose oxaliplatin was administered on day 0 and 7 (Fig. 8A). Consistent with the phenotype observed in *Ido1*[-/-] mice, Epacadostat treatment exhibited a significant protection against oxaliplatin-induced toxicity. Compared with control mice, Epacadostat-treated mice exhibited markedly reduced severity of toxicity symptoms (Fig. 8B–E and Fig. S7A). Histopathological assessment of intestinal tissues revealed substantial improvement in the Epacadostat group compared with the vehicle group (Fig. 8F–I). Collectively, these results demonstrate that pharmacological inhibition of IDO1 with Epacadostat could attenuate chemotherapy-induced toxicity.

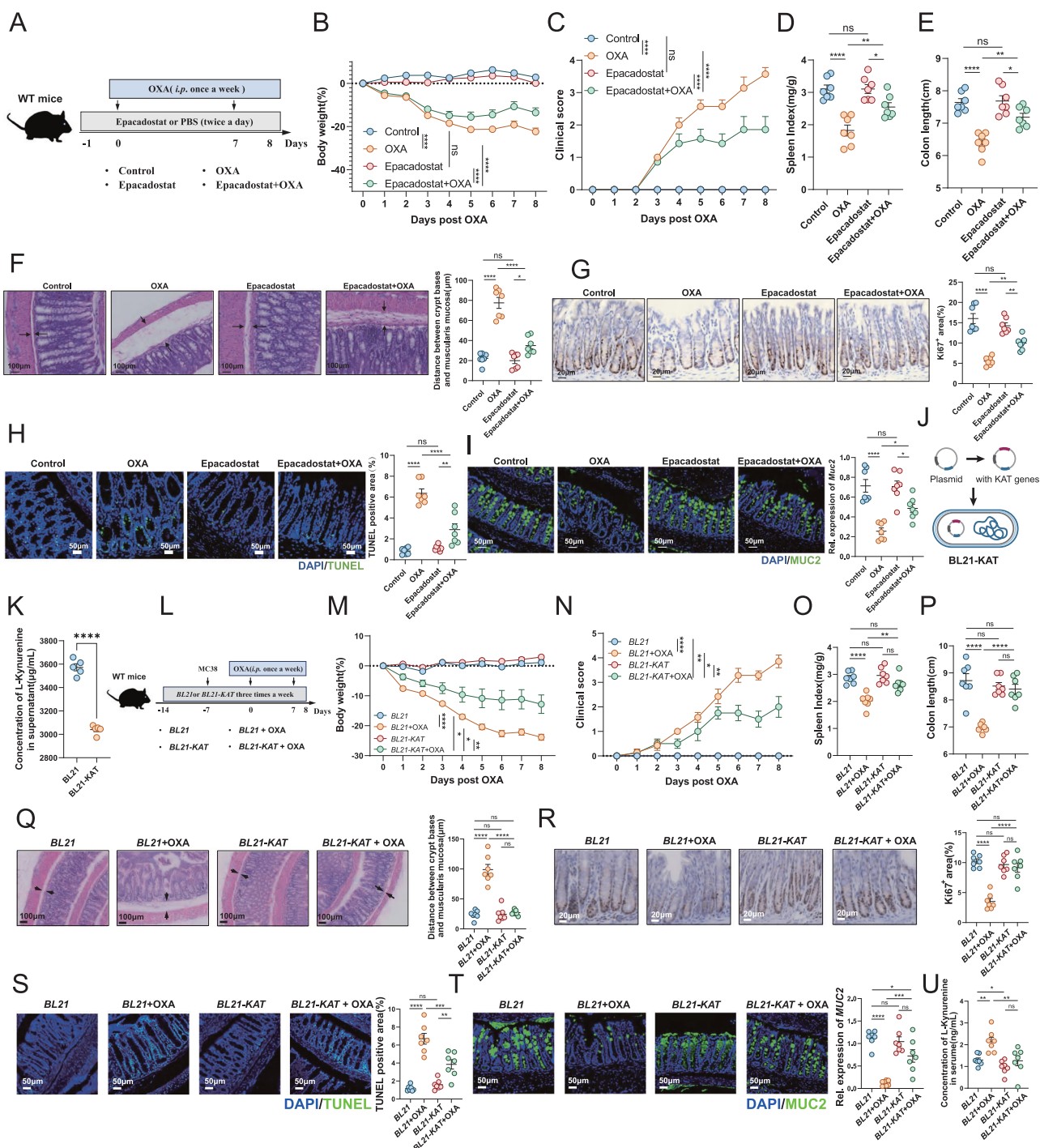

While suppressing L-kynurenine biosynthesis in host cells is effective, we hypothesized that directly enhancing its metabolism in the gut lumen could offer a complementary strategy. We established engineered bacteria by using *Escherichia coli* strain *BL21*, which over-expresses the downstream enzyme kynurenine aminotransferase (KAT) (*BL21-KAT*) (Fig. 8J). A significantly lower level of L-kynurenine was found in the supernatant from *BL21-KAT* (Fig. 8K). To evaluate the therapeutic potential of *BL21-KAT* in oxaliplatin-induced toxicity, we established a subcutaneous tumor xenograft model in which mice were gavaged with *BL21-KAT* three times a week followed by oxaliplatin treatment once a week (Fig. 8L). Similar tumor size and tumor weight were observed in mice with *BL21-KAT* or *BL21* after oxaliplatin intervention, indicating that *BL21-KAT* intervention did not influence the antitumor efficacy of oxaliplatin (Fig. S7B, C). Moreover, compared

with mice treated with *BL21* and oxaliplatin, mice treated with *BL21-KAT* and oxaliplatin exhibited significant improvements in weight loss, clinical score, spleen index and colon length (Fig. 8M–P and Fig. S7D). Histopathological analysis also revealed a significant alleviation of oxaliplatin-induced intestinal toxicity in mice treated with *BL21-KAT* and oxaliplatin (Fig. 8Q–T). More importantly, the concentration of L-kynurenine in the plasma of mice treated with *BL21-KAT* and oxaliplatin was significantly lower than that in the plasma of mice treated with *BL21* and oxaliplatin, whereas similar concentrations of L-kynurenine were observed between mice treated with *BL21-KAT* alone and those treated with *BL21-KAT* and oxaliplatin (Fig. 8U). Therapeutic targeting of the IDO1–L-kynurenine axis, either through pharmacological inhibition (Epacadostat) or engineered L-kynurenine-metabolizing probiotics, significantly reduces chemotherapy-induced

**Fig. 8 | Targeting the IDO1–L-kynurenine axis with pharmacological and microbial strategies abrogates chemotherapy-induced toxicity. A** Schematic diagram showing the process of oxaliplatin toxicity mice model treated with OXA or Epacadostat. Changes of body weight (**B**), clinical score (**C**), spleen index (**D**), and colon length (**E**). **F** Representative histopathological images of colon (left) and quantification for the gaps between crypt bases and muscularis mucosa (right). Arrows indicate gaps between crypt bases and muscularis mucosa. Scale bar = 100 μm. **G** Immunohistochemical staining of Ki67 in intestinal tissue. Representative images (left) and the quantification of Ki67 area (right). Scale bar = 20 μm. **H** Immunofluorescence analysis of TUNEL in intestinal tissues. Representative images (left) and the quantification of TUNEL positive area (right). Scale bar = 50 μm. **I** Immunofluorescence analysis and transcript expression of MUC2 in intestinal tissues. Representative images (left) and the quantification of MUC2 expression (right). Scale bar = 50 μm. **B–I** n = 7 mice per group. **J** Experiment workflow of the engineering bacteria construction. **K** Concentration of L-kynurenine in the supernatant of *BL21* and *BL21-KAT*. (n = 5 for both groups). **L** Schematic diagram showing the process of subcutaneous tumor xenograft model treated with *BL21*, *BL21* + OXA, *BL21-KAT* or *BL21-KAT* + OXA. Changes of

body weight (**M**), clinical score (**N**), spleen index (**O**) and colon length (**P**). **Q** Representative histopathological images of colon (left) and quantification for the gaps between crypt bases and muscularis mucosa (right). Arrows indicate gaps between crypt bases and muscularis mucosa. Scale bar = 100 μm. **R** Immunohistochemical staining of Ki67 in intestinal tissue. Representative images (left) and the quantification of Ki67 area (right). Scale bar = 20 μm. **S** Immunofluorescence analysis of TUNEL in intestinal tissues. Representative images (left) and the quantification of TUNEL positive area (right). Scale bar = 50 μm. **T** Immunofluorescence analysis and transcript expression of MUC2 in intestinal tissues. Representative images (left) and the quantification of MUC2 expression (right). Scale bar = 50 μm. **U** Concentration of L-kynurenine in the serum from mice treated with of *BL21*, *BL21* + OXA, *BL21-KAT* or *BL21-KAT* + OXA. **M–U** n = 7 mice per group. Quantitative data are expressed as the mean ± standard error of the mean (S.E.M). \*$P < 0.05$, \*\*$P < 0.01$, \*\*\*$P < 0.001$, \*\*\*\*$P < 0.0001$; ns, not significant. *P* values were determined by the unpaired two-tailed Student's t-tests (**K**), two-way ANOVA with Tukey's post hoc test (**B**, **C**, **M**, **N**) and one-way ANOVA with Tukey's post hoc test (**D–I**, **O–U**). Source data are provided as a Source Data file.

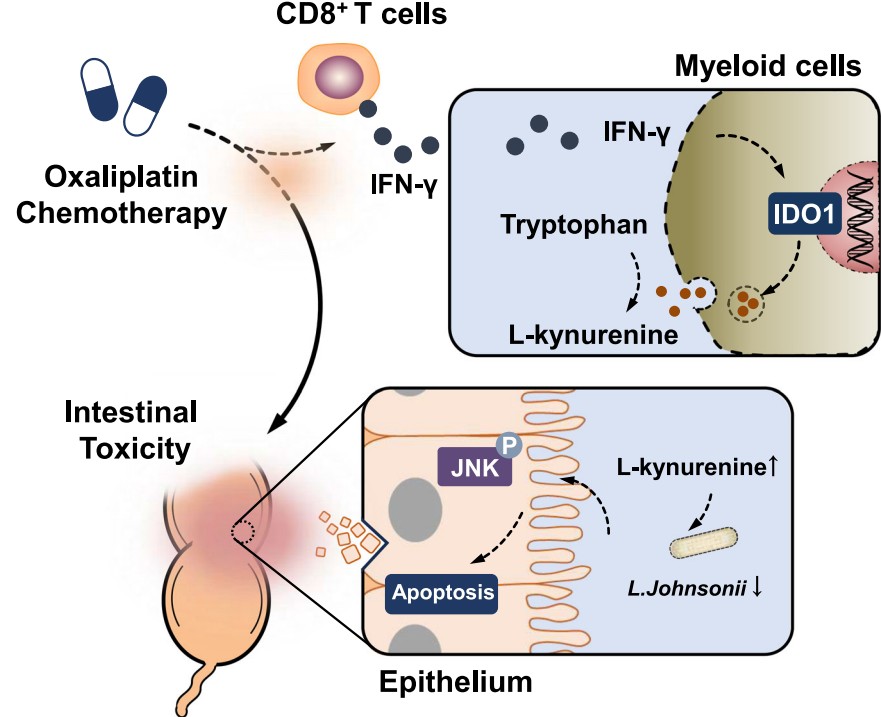

**Fig. 9 | Schematic diagram of the mechanism: the L-kynurenine–myeloid cell–microbiota axis driving chemotherapy-induced intestinal toxicity.** Intestinal toxicity induced by oxaliplatin chemotherapy is driven by CD8+ T cell-derived IFNγ, which activates the IDO1 pathway in myeloid cells and elevates L-kynurenine

levels. This surge in L-kynurenine causes gut dysbiosis (loss of *L. johnsonii*), which ultimately activates the TNFα/JNK signaling cascade, resulting in epithelial cell apoptosis and intestinal damage.

toxicity, offering clinically actionable strategies to improve patient tolerance.

## Discussion

Collectively, the results of our study demonstrate that chemotherapy-induced IFNγ production potently induces IDO1 expression, driving L-kynurenine accumulation and gut microbiota dysbiosis. These changes activate the TNFα/JNK proapoptotic pathway, culminating in intestinal toxicity. Crucially, downregulating IDO1 in myeloid cells reduced L-kynurenine levels, enabling *L. johnsonii* expansion and subsequent suppression of TNFα/JNK signaling (Fig. 9). Our findings establish targeted inhibition of the IDO1–L-kynurenine axis as a therapeutic strategy and mitigate chemotherapy-induced tissue damage.

Our study demonstrated that IFNγ plays a crucial role in IDO1 expression. These findings are consistent with a substantial body of

literature that identifies IFNγ as a primary inducer of IDO1 in various cell types, establishing a direct link between this cytokine and the immunosuppressive enzyme IDO1[24,25]. The molecular mechanism underlying this observation is well-established and primarily involves the JAK-STAT signaling pathway. For instance, Nami Yamashita et al. demonstrated that MUC1-C directly activates the inflammatory IFNγ-driven JAK1/STAT1/IRF1 pathway, which in turn induces the expression of key immunosuppressive effectors, including IDO1[26]. Collectively, our findings corroborate IFNγ as a pivotal upstream regulator of IDO1 expression across biological contexts.

L-kynurenine is a downstream metabolite of tryptophan via the key rate-limiting enzyme IDO1[27]. A previous study demonstrated that cisplatin modulated the metabolism of L-kynurenine by increasing the activity of the enzyme IDO1[28]. Application of 5-FU had also found to enhance the synthesis of ROS and subsequently upregulate the IDO1

expression, resulting in the production of L-kynurenine[29]. More importantly, higher concentration of L-kynurenine was found to be associated with intestinal inflammation in different clinical cohorts[30]. Similarly, our current data demonstrated that high-dose oxaliplatin upregulated L-kynurenine expression. High levels of L-kynurenine were associated with the occurrence of chemotherapy-induced side effects, especially the development of intestinal toxicity. Interestingly, a previous study demonstrated that targeting the synthesis of L-kynurenine could downregulate the production of proinflammatory cytokines in macrophages[31]. Our data also emphasized the role of myeloid cells in the L-kynurenine biosynthesis during the chemotherapy-induced toxicity. These evidences suggest the chemotherapy enhance the L-kynurenine metabolism in myeloid cells to promote the development of chemotherapy-associated toxicity.

It is well known that the secretion of L-kynurenine can mediate the transition of immune tolerance, indicating that L-kynurenine can indirectly mediate intestinal toxicity. Several studies have indicated a potential association between chemotherapy-induced toxicity and the gut microbiota. Changes in the microbial composition induced by chemotherapy profoundly influence intestinal injury[32]. Microbial derived β-glucuronidase actively contributes to irinotecan-induced toxicity in the gastrointestinal tract[33]. He et al. further revealed that depletion of fecal *Lactobacillus* and *Bifidobacterium* could exacerbate the chemotherapeutic toxicity through the suppression of interleukin-10 from macrophages[8]. Depletion of probiotics was also associated with chemotherapy-induced gastrointestinal toxicity in lymphoblastic leukemia patients[10]. In this study, we observed a specific accumulation of *L.johnsonii* in *Ido1⁻/⁻* mice treated with high-dose oxaliplatin, while higher level of L-kynurenine could downregulate the colonization of *L.johnsonii*. Therefore, we demonstrated that higher level of L-kynurenine mediated the chemotherapy-induced toxicity through a downregulation of *L.johnsonii*.

The interaction between the alteration of host metabolism and the shift in the gut microbiota in the context of chemotherapy-induced intestinal toxicity is complicated. A recent study demonstrated that chemotherapy-induced epithelial apoptosis and the release of purine-containing metabolites from dying cells drive the transcriptional rewiring of *Enterobacteriaceae*, including fundamental shifts in bacterial respiration and the promotion of purine utilization-dependent expansion[34]. In our current study, how L-kynurenine modulates the colonization of *L. johnsonii* is an interesting topic. The upregulation of L-kynurenine strongly affected intestinal pH, intestinal barrier function and the immune response[35]. Alteration of the intestinal homeostasis by these factors may further reduce the colonization of probiotics[36]. Our own in vitro experiments provide direct evidence for this hypothesis, showing that the addition of L-kynurenine to the culture medium significantly inhibited the growth of *L. johnsonii*. However, further studies were encouraged to clarify the underlying mechanism between L-kynurenine and *L.johnsonii*.

L-Kynurenine not only modulates the gut microbiota by directly influencing the microenvironment but also indirectly exacerbates gut microbial dysbiosis by activating specific host signaling pathways[37]. Evidence suggests that the excessive accumulation of L-kynurenine can activate TNF-α signaling pathway[38]. As a proinflammatory cytokine, TNF-α can further activate the downstream JNK signaling pathway[39]. Once activated, the JNK pathway can subsequently leads to apoptosis of intestinal epithelial cells[40]. This excessive apoptosis compromises the integrity of the intestinal mucosal barrier and increases gut permeability. Studies have shown that maternal consumption of a fermented diet enriches the *Lactobacillus* genus, which effectively inhibits the activation of the JNK signaling pathway in the offspring's intestines, thus reducing intestinal epithelial cell apoptosis and ultimately alleviating intestinal inflammation[41]. In this inflammatory setting, the ecological niche for beneficial bacteria such as *Lactobacillus* is compressed, while opportunistic pathogens can proliferate, leading to severe gut microbial dysbiosis[42]. Therefore, L-kynurenine-induced gut dysbiosis contributes to intestinal epithelial cell damage, creating a pathological microenvironment that perpetuates microbial imbalance.

Collectively, the results of our present study highlight the important role of L-kynurenine in chemotherapy-induced toxicity. The downregulation of L-kynurenine from myeloid cells enhanced the accumulation of *L. johnsonii* to improve intestinal toxicity. Intervention targeting L-kynurenine metabolism may be a potential therapeutic strategy to improve chemotherapy tolerance.

## Methods

### Human samples

A cohort of colorectal cancer (CRC) patients treated with oxaliplatin-based chemotherapy was recruited from The Sixth Affiliated Hospital of Sun Yat-sen University. Sixty four patients were recruited and stratified into higher toxicity group (*n* = 30) and lower toxicity group (*n* = 34) based on the National Cancer Institute's Common Terminology Criteria for Adverse Events (NCI-CTCAE v5.0). Demographic and clinical characteristics of the cohort are summarized in Table S1. While an independent validation cohort of 28 patients was additionally collected to verify the findings. The study was approved by the Human Medical Ethics Committee of the Sixth Affiliated Hospital of Sun Yat-sen University. Written informed consent was obtained from all participants prior to their inclusion in the study. The approve number of this study was 2020ZSLYEC-266. Serum samples were collected after chemotherapy and stored at −80 °C.

### Mice

Six- to eight-week-old male mice on the C57BL/6J (WT) background were used. C57BL/6 J WT mice, *Ido1⁻/⁻* mice, *Ido1flox/flox* mice, Lyz2-Cre mice and Villin-Cre mice were obtained from GemPharmatech company. Myeloid-specific (*Ido1fl/fl Lyz2*-Cre) and intestinal epithelial-specific (*Ido1fl/fl Villin*-Cre) *Ido1* knockout mice were generated through strategic cross-breeding of the respective Cre and floxed lines. All animals were maintained under specific pathogen-free (SPF) conditions with a 12-hour light/dark cycle and ad libitum access to food and water at the Sixth Affiliated Hospital of Sun Yat-sen University and Guangzhou Ruige Biological Technology Co., Ltd. Experimental procedures were strictly conducted following protocols approved by the Institutional Animal Care and Use Committee (IACUC). The approve number of this study were IACUC-2023060701 and 20220801-001.

### Cell culture

The cell lines were cultured in Gibco Dulbecco's Modified Eagle Medium (DMEM) or RPMI-1640 medium with 10% fetal bovine serum (GIBCO) and 1% penicillin – streptomycin (GIBCO) at 37 °C in 5% CO2 and 95% humidity.

### Bacteria culture

Media for cultivation of organisms were listed in the Method Details were pre-reduced in the aerobic chamber (90% N2, 5% CO2, 5% H2) for 48 h before inoculation. Bacterial cultures intended for in vitro and in vivo assays were grown overnight at 37 °C in the aerobic or anaerobic chamber (CAW1100, Guangzhou Huanghe Instrument Technology Co., Ltd.). Bacterial isolation and identification were achieved by colony picker (RapidPick SP, GuangZhou Jinke Chian Technology Co., Ltd.), detection system (CMI-1600, Guangzhou Chengyi Imp & Exp Co., Ltd.) and dilution system (Easy spiral dilute, Zhongke Scientific & Technical Co., Ltd.).

### Oxaliplatin toxicity model

A toxic dose of oxaliplatin (20 mg/kg body weight) or PBS control was administered to mice via peritoneal injection once a week. The mice were then housed in sterile autoclaved cages and provided

standard chow and water ad libitum, unless otherwise noted. Body weight, clinical scores (determined using a cumulative scoring system, based on weight loss, temperature changes, physical appearance, posture, and mobility), and survival were monitored. To evaluate the role of L-kynurenine, mice were gavaged with L-kynurenine (20 mg/kg body weight) or PBS control three times a week. To evaluate the role of IDO1, *Ido1* [−/−] mice, *Ido1*[flox/flox] *Lyz2-Cre* mice and *Ido1*[flox/flox] *Villin-Cre* mice were established for oxaliplatin toxicity model as described above. To evaluate the role of microbiota, mice were treated with broad-spectrum antibiotic cocktail (ampicillin 0.2 g/L, metronidazole 0.2 g/L, neomycin 0.2 g/L, vancomycin 0.1 g/L) for one week before oxaliplatin treatment and continued until the end of the experiment. In accordance with the protocols approved by IACUC, the maximal allowable body weight loss was 30% relative to the initial weight.

### Subcutaneous tumor xenograft experiment

To establish a subcutaneous tumor xenograft model, CRC cell line MC38 ($1 \times 10^6$ /100ul) were injected into the loose subcutaneous tissues of the mice right back. After 7 days, C57BL/6 J WT mice were treated with oxaliplatin (20 mg/kg body weight) intraperitoneally once a week. To evaluate the role of *L. johnsonii*, mice were gavaged with *L. johnsonii* ($1 \times 10^9$ CFU/dose) three times a week for one week before tumor injection and continued until the end of the experiment. To evaluate the role of IDO1, *Ido1*[flox/flox] *Lyz2-Cre* mice was established for subcutaneous tumor xenograft model as described above. To evaluate the role of engineered bacteria, mice were gavaged with *BL21-KAT* or *BL21* ($1 \times 10^9$ CFU/dose) three times a week for one week before tumor injection and continued until the end of the experiment. In accordance with the protocols approved by the IACUC, the maximal permitted tumor volume was 2000 mm$^3$.

### Fecal microbiota transplantation

Feces from WT C57BL/6 J and *Ido1*[−/−] donor mice were collected in 20% autoclaved glycerol and quickly suspended and homogenized. After centrifugation at $100 \times g$ at 4 °C for 30 s, the supernatants were collected. Recipient mice were treated with a broad-spectrum antibiotic cocktail for one week and subsequently were gavaged with 0.2 mL fecal suspension three times a week. Subsequently, mice were challenged with oxaliplatin once a week at 1 week post-transplantation.

### IDO1 inhibitor animal experiment

Mice were administered Epacadostat (50 mg/kg) or PBS (control) via oral gavage twice daily until the end of the experiment. Following the initiation of Epacadostat or PBS treatment, the mice received oxaliplatin or PBS control therapy. A toxic dose of oxaliplatin (20 mg/kg body weight) or PBS was administered once per week via intraperitoneal injection.

### In vivo IFNγ neutralization

To assess the role of IFNγ, mice were injected intraperitoneally on days −1, 2, 5, and 8, with 300 μg anti-IFNγ (clone R4−6A2; BioXCell) or isotype rat IgG1 controls (clone HRPN; BioXCell). Following the initiation of anti-IFNγ treatment, the mice received oxaliplatin therapy. A toxic dose of oxaliplatin (20 mg/kg body weight) or PBS was administered once per week via intraperitoneal injection.

### Isolation and culture of *Lactobacillus johnsonii*

*L. johnsonii* was isolated from healthy individual and verified using 16S rRNA sequencing (V4 sequences). The isolation of *L. johnsonii* using an automated colony picker (RapidPick SP, GuangZhou Jinke Chian Technology Co., Ltd.), detection system (CMI-1600, Guangzhou Chengyi Imp & Exp Co., Ltd.) and dilution system (Easy spiral dilute, Zhongke Scientific & Technical Co., Ltd.). *L. johnsonii* was cultured in

MRS medium at 37 °C in an anaerobic chamber using gas mix consisting of 5% hydrogen, 10% carbon dioxide and 85% nitrogen for 24 h.

### Histopathological analysis of spleen and colon

Spleens and Swiss-rolled like colon tissues were freshly collected and immersed in 10% neutral buffered formalin. The tissue was then embedded in paraffin and sliced into 5 mm sections, which were sequentially performed following standard procedures. The pathological changes in spleens and colon were quantified by hematoxylin and eosin(H&E) staining.

### Enzyme-linked immunosorbent assay (ELISA)

The concentrations of IFN-γ in serum and fecal samples were quantified using a commercial sandwich enzyme-linked immunosorbent assay (ELISA) kit (Telenbiotech, TL-E1382), according to the manufacturer's instructions. Briefly, serum samples were diluted and fecal samples were homogenized in PBS followed by centrifugation to obtain supernatants. Standards and samples were incubated in antibody-precoated plates. After a series of incubations with a biotinylated detection antibody and streptavidin-HRP conjugate, the reaction was developed with tetramethylbenzidine (TMB) substrate and stopped with acid. The absorbance was measured at 450 nm. The cytokine concentration was determined by interpolating values from the standard curve. For fecal samples, results were normalized to the initial wet weight of the sample.

### Immunohistochemistry (IHC)

Paraffin-embedded tissue sections (4 μm) were deparaffinized in xylene, rehydrated through graded ethanol, and subjected to heat-induced antigen retrieval in sodium citrate buffer (10 mM, pH = 6.0) at 95 °C for 10 min. Endogenous peroxidase activity was quenched with 3% H$^2$O$^2$, followed by blocking with 5% BSA in PBS for 30 min. Sections were incubated overnight at 4 °C with anti-Ki67 antibody (#GB121141,Servicebio, 1:250 dilution). Next, washed with PBST, and treated with HRP-conjugated secondary antibody (#G1214, servicebio) for 30 min at room temperature in the dark. Signal was visualized using DAB chromogen, followed by counterstaining with hematoxylin for 1 min, and mounted with resin. The slides were scanned by slide scanning system (Shengqiang, China). The Ki-67 positive area was evaluated by ImageJ.

### Immunofluorescence staining

The tissues sections were freshly isolated and fixed with 10% formalin before embedding in paraffin wax. The whole staining was performed on paraffin-embedded sections (4 mm). After deparaffination with dimethylbenzene and rehydration with ethanol, the slides were immersed in boiling sodium citrate buffer (10 mM, pH = 6.0) for 10 min to retrieve antigen. Antigen retrieval was performed by boiling slides in 10 mM sodium citrate buffer (pH 6.0) for 10 min, followed by cooling to room temperature. Sections were permeabilized and blocked with 0.6% Triton X-100 in PBS for 15 min and washed three times with PBST (PBS containing 0.1% Tween-20, 5 min per wash). The tissues samples were immune-stained with primary antibody(#GB120002, servicebio, 1:250 dilution) by incubating overnight at 4 C. After three PBST washes, sections were incubated with secondary antibody (#ab150077, Abcam, Alexa Fluor 488) for 30 min at room temperature in the dark. Nuclei were counterstained with DAPI, and slides were cover slipped and sealed. Then examined slides under a laser scanning confocal microscope (Zeiss, Germany).

### TdT-mediated dUTP nick-end labeling (TUNEL) assay

Tissue apoptosis was detected with an in-situ cell death detection kit (#101-131, Goonie, China) following the manufacture's instruction. Briefly, paraffin-embedded intestinal sections were dewaxed in xylene

and rehydrated through gradient alcohols. The slides were then incubated with 20ug/mL proteinase K for 5 min followed by PBS washing. Next, the samples were incubated with the TUNEL reaction mixture in a humidified chamber at room temperature for 60 min avoid light. This was followed by PBS washes and incubation with 1 mg/mL DAPI. After staining, the sections were scanned under a laser scanning confocal microscope (Zeiss, Germany). Quantification was performed with Image J.

### RNA and DNA extraction and quantitative real-time PCR

Quantitative real-time PCR RNA extraction was performed using the FastPure Cell/Tissue Total RNA Isolation Kit according to the manufacturer instructions, and the HiScript III 1st Strand cDNA Synthesis Kit with a gDNA wiper was used to reverse transcribe the RNA into cDNA. Fecal or bacterial DNA was obtained using an AmPure Microbial DNA Kit (D7111, Megan). qPCR was performed using the SYBR Green Supermix Kit on the QuantStudioTM 7 Flex Real-Time PCR System (Thermo Fisher Scientific). Primer sequences used in this study was shown in Table S2.

### Heterologous expression of *KAT* in *E. coli*

The kynurenine aminotransferase (KAT) expression system was established as previously described[30]. Briefly, the KAT open reading frame (ORF) (Table S4) was codon-optimized for *E. coli* expression using the OPTIMIZER online tool, OptimumGene™ (Genscript), and manual adjustments to eliminate rare codons, optimize GC content, and remove negative CIS elements or repeats. The optimized sequence was cloned into the pET15b vector. For protein overexpression, the pET15b-KAT plasmid and an empty pET15b control were transformed into chemically competent *E. coli BL21* (DE3) via standard heat-shock to generate *E. coli BL21-KAT* and *E. coli BL21-WT*, respectively. Transformants were selected on ampicillin plates (100 mg/mL) at 37 °C. A single colony was expanded overnight and inoculated into 50 mL of LB medium. Once the culture reached an OD600 of 0.6-0.8, KAT expression was induced with 1 mM IPTG at 37 °C under agitation (220 rpm).

### Cell isolation of spleen mononuclear cells

Spleens were completely isolated from mice and crushed with forceps, and single cells were isolated in PBS using a 70-μm cell strainer. The cells were washed with 1x PBS and centrifuged (100 × g for 5 min), and then red blood cell lysis containing splenocytes was pipetted up. The culture medium was then added to the cells and centrifuged at 100 × g at 4 °C for 5 min. Single-cell suspensions were diluted in Roswell Park Memorial Institute (RPMI) medium.

### Cell culture and cellular stimulation

The murine epithelial cell line NCM460 was purchased from the American Type Culture Collection and cultured at 37 °C in DMEM (Gibco) supplemented with 10% FBS (Gibco) in a 5% $CO_2$ atmosphere. 5 ×10⁵ cells were grown in a 6-well plate and co-cultured with bacteria, L-kynurenine, or oxaliplatin for 24 h. Subsequently, protein was extracted for Western blot analysis or cells were collected for apoptosis analysis by flow cytometry.

### Flow cytometry analysis

The effectors IFN-g (BD Biosciences, 562303) were detected by adding Activation Cocktails (BioLegend, 423303) to 1 ml of cell suspension and incubating for 6 h at 37 °C. Then, 1 × 10⁶ cells were stained with Fixable Viability Stain 700 (BD Biosciences, 564997) in serum-free buffer for 30 min. Then, cells were incubated with anti-CD16/32 antibody (BioLegend,101320) for 10 min to block non-specific binding of immunoglobulin to the Fc receptors. Afterwards, the appropriate amount of pre-diluted fluorescent labeled antibody (BD Biosciences, 563024, 553051, 552877, 557659, 564143; STARTER, S0B5093) was

added to each tube as recommended by the manufacturer. Cells were incubated in the 4 °C refrigerator for 30 min and fixed with Fixation Buffer (BD Biosciences, 562574) for 30 min at room temperature. After washing, the membranes were ruptured (BD Biosciences, 562574) and intracellular fluorescent antibodies were added to each tube and incubated for 30 min at room temperature. Prepared cells were analyzed by flow cytometry (Beckman Coulter). The gating strategy is illustrated in the Fig. S8.

### Cell apoptosis assay

2 × 10⁵ of cells were seeded into a 6-well plate, and then treated with bacteria, L-kynurenine, or oxaliplatin for 24 h. Cell apoptosis was detected by Apoptosis Detection Kit (Multi sciences, AP107). Prepared cells were analyzed by flow cytometry (Beckman Coulter).

### Western blot

Total protein was extracted using RIPA lysis buffer supplemented with protease inhibitors and phosphatase inhibitors (Servicebio, Wuhan, China) followed by sonication and centrifugation at 4000 rpm for 15 min. Then, Protein samples (20 μg) were separated by Bis/Tris-polyacrylamide gel electrophoresis and transferred onto PVDF membranes (Millipore, Shanghai, China). Membranes were blocked in 5% skimmed milk and incubated with primary antibodies and secondary antibodies. The protein bands were visualized using Enhanced Chemiluminescence (ECL) Reagent (Millipore, 2236001). Protein signals were captured and imaged using the Azure 280 Imaging System (Azure Biosystems, USA).

### Targeted metabolomics for tryptophan metabolism

**Chemicals and reagents.** Serum tryptophan and its metabolites were quantified by Maiwei Metabolic Biotechnology Co., Ltd. (Wuhan, China). HPLC-grade methanol (MeOH) and acetonitrile (ACN) were obtained from Merck (Darmstadt, Germany), while acetic acid and all analytical standards were sourced from Sigma-Aldrich or OlChemim (Olomouc, Czech Republic). Ultrapure water was prepared using a Milli-Q system (Millipore, USA). Individual standard stock solutions (1 mg/mL in MeOH) were stored at −20 °C and further diluted with MeOH to prepare working solutions.

**Sample preparation.** Briefly, 50 mg of thawed serum was extracted with 500 μL of MeOH, supplemented with 20 μL of an internal standard (IS) mixture (250 ng/mL). The mixture was vortexed for 3 min, incubated at −20 °C for 30 min, and centrifuged at 3000 × g for 10 min at 4 °C. A 250 μL aliquot of the supernatant was collected and subjected to a second centrifugation (3000 × g for 5 min at 4 °C). Finally, 150 μL of the resulting supernatant was recovered for LC-MS/MS analysis.

**LC-MS/MS conditions.** Metabolomic profiling was performed using an ExionLC AD UPLC system coupled with a QTRAP 6500+ mass spectrometer (Sciex). Chromatographic separation was achieved on a Waters ACQUITY UPLC HSS T3 C18 column (100 mm × 2.1 mm i.d.,1.8 μm) maintained at 40 °C. The mobile phase consisted of 0.1% formic acid in water (A) and 0.1% formic acid in acetonitrile (B). The gradient program was as follows: 0-1 min, 10% B; 1-8 min, 10-95% B; 8-9.5 min, 95% B; and 9.6-12 min, 10% B. The flow rate was 0.35 mL/min with a 5 μL injection volume.

The mass spectrometer, equipped with an ESI Turbo Ion-Spray interface, operated in both positive and negative ionization modes. Key source parameters included: source temperature, 550 °C; ion spray voltage (IS) 5500 V (Positive),−4500 V(Negative); curtain gas (CUR) was set at 35 psi. Metabolites were quantified using scheduled multiple reaction monitoring (MRM). Data acquisition and quantification were conducted using Analyst 1.6.3 and MultiQuant 3.0.3 software (Sciex), respectively.

**Metabolomics analysis.** The Wilcoxon rank-sum test was used to identify concentrations that differed in abundance between higher and lower toxicity group. The linear mixed effects modeling with the R package MaAsLin2 to perform the association of each clinical parameter and specific metabolites. The p values and beta coefficients generated from this model indicate the significance and effect estimates of examining whether the concentration of a specific metabolite changes across the change of clinical parameter in a dose-response manner.

**Liquid chromatograph mass spectrometer (LC-MS) analysis**
For the serum samples 100 μL samples were mixed with 900 μL mass spectrometry grade pre-chilled methanol, then vortexed for 5 min. The mixture was then centrifuged at 3000x g and 4 °C for 10 minutes, and the supernatant was collected. An Agilent 1290 Infinity II liquid chromatography system coupled to an Agilent 6495 A triple quadrupole LC-MS system was used to quantitate L-Kyn. Data were collected by Mass Hunter workstation. For data processing, the peak area was integrated by use of Agilent Mass Hunter Vista Flux software, and Agilent Metabolite ID software.

**16S ribosomal RNA gene sequencing and analysis**
DNA from frozen stool samples was extracted using a FastDNA Spin Kit for Soil. For 16S rRNA gene sequencing, the V3-V4 variable region was amplified using 2-step PCR. In the first step, 10 ng genomic DNA was used as a template for the first PCR with a total volume of 20 μl using the 338 F (5′ACTCCTACGGGAGGCAGCAG-3′) and 806 R (5′-GGAC-TACHVGGGTWTCTAAT-3′) primers appended with Illumina adaptor sequences. The amplicons were purified, checked on a fragment analyzer, quantified, followed by equimolar multiplexing, and sequenced on an Illumina MiSeq PE300 platform.

**Single cell RNA sequencing**
**Tissues dissociation and preparation of mice intestinal tissues.** Intestinal tissues were harvested from *Ido1^{-/-}* mice and their corresponding littermate controls. To minimize environmental bias, all mice were co-housed from birth and received no prior experimental treatment. Single-cell RNA sequencing (scRNA-seq) was performed by LC-Bio Technologies Co., Ltd. (Hangzhou, China).

**Pre-processing.** Fresh tissues were placed in sterile, RNase-free dishes containing ice-cold, calcium/magnesium-free 1 × PBS. Tissues were minced into 0.5 mm² fragments, washed with PBS, and meticulously cleared of blood stains and mesenteric fat.

**Enzymatic digestion.** Tissue fragments were dissociated into single cells using a digestion cocktail (0.35% collagenase IV5, 2 mg/ml papain, 120 Units/ml DNase I) in a in 37 °C water bath with shaking for 20 min at 100 × g.

**Cell recovery.** The reaction was neutralized with 1 × PBS supplemented with 10% (v/v) fetal bovine serum (FBS). After mechanical trituration using a Pasteur pipette, the suspension was filtered through a 70 μm cell strainer and centrifuged at 300 g for 5 min at 4 °C.

**Purification.** The cell pellet was resuspended in 100ul 1× PBS (0.04% BSA). Residual red blood cells were removed using 1× RBC Lysis Buffer (MACS 130-094-183), followed by dead cell removal using the Miltenyi Dead Cell Removal Kit (MACS 130-090-101).

**Quality control.** The final pellet was washed twice and resuspended in 50 μl of 1× PBS (0.04% BSA). Cell viability was assessed via trypan blue exclusion (threshold >85%). Cells were counted using a Countess II Automated Cell Counter and adjusted to a final concentration of 700–1200 cells/μl.

**Chromium 10x Genomics library and sequencing.** Single-cell Gel Bead-in-Emulsions (GEMs) were generated by loading cellular suspensions onto a 10x Genomics GemCode instrument. Libraries were constructed using the Chromium Next GEM Single Cell 3′ Reagent Kits v3.1. Full-length barcoded cDNAs were amplified via PCR. R1 sequences were introduced during GEM incubation, while P5, P7, sample indices, and R2 sequences were added during library construction via end repair, A-tailing, and adapter ligation. The final Illumina-ready libraries contained 16 bp 10x Barcodes and 10 bp UMIs in Read 1, with cDNA fragments sequenced in Read 2.

**Data quality control.** Raw data were processed using **Cell Ranger (v3.1.0)** for FASTQ conversion, genome alignment, and UMI counting. Low-quality barcodes were filtered, and valid cell droplets were identified using the EmptyDrops method. Cell-by-gene matrices were generated via UMI counting and cell barcode calling. Downstream analyses were performed by Seurat v5.0[43]. Cells with mitochondrial gene≥25%, Hemoglobin gene≥3%, n.features<200 genes, or n.count>6000 genes were filtered out. Doublets were identified and removed using DoubletFinder (v2.0.4). Through the quality control steps above, low-quality cells, droplets with ambient RNA, and suspicious doublets were removed. Samples were integrated and batch effects were minimized with canonical correlation analysis (CCA).

**Unsupervised clustering and markers identification.** Data were log-normalized and scaled. After PCA-based linear dimensional reduction on highly variable genes, cells were clustered and visualized using UMAP. Cluster-specific markers were identified using the FindMarkers function (Wilcoxon rank-sum test). Differentially expressed genes (DEGs) were defined by >10% cells in one of the two conditions, with |log2FC| > 0.5, and adjusted p value < 0.05 (Bonferroni correction) are defined as differentially expressed.

**Cell-cell interaction analysis.** Potential cell-cell interactions between different immune cells were identified using CellChat v1.1.3[44]. Communication probabilities were calculated by integrating gene expression with the CellChatDB database of ligand-receptor pairs. The aggregated signaling network was visualized using the netVisual_bubble function to highlight significant interaction strengths between immune cell subsets.

**Bacterial strand-specific RNA-seq**
**RNA extraction.** Total RNA was extracted from the tissue using CTAB method and genomic DNA was removed. Only high-quality RNA sample was used to construct sequencing library.

**Library construction and sequencing.** Ribosomal RNA (rRNA) depletion instead of poly(A) purification is performed by RiboCop rRNA Depletion Kit for Mixed Bacterial Samples (lexogen, USA) and then all mRNAs were broken into short (200nt) fragments by adding fragmentation buffer firstly. Secondly double-stranded cDNA was synthesized with random hexamer primers (Illumina). When the second strand cDNA was synthesized, dUTP was incorporated in place of dTTP. Then the synthesized cDNA was subjected to end-repair, phosphorylation and 'A' base addition according to Illumina's library construction protocol. RNA-seq transcriptome library was prepared following Illumina® Stranded mRNA Prep, Ligation (San Diego, CA) using of total RNA. paired-end RNA-seq library was sequenced with the Illumina Novaseq Xplus(or other new sequenator) (Illumina Inc., San Diego, CA, USA).The processing of original images to sequences, base-calling, and quality value calculations. The clean reads by removing low-quality sequences, reads with more than 10% of N bases (unknown bases) and reads containing adaptor sequences.

**Expression analysis.** Quantify gene and transcript abundances from single-end or paired-end RNA-Seq data using RSEM, RSEM computes maximum likelihood abundance estimates using the Expectation Maximization (EM) algorithm for its statistical model, including the modeling of paired-end (PE) and variable-length reads, fragment length distributions, and quality scores, to determine which transcripts are isoforms of the same gene. The FPKM and TPM are able to eliminate the influence of different gene length and sequencing discrepancy on the calculation of gene expression. Therefore, the calculated gene expression can be directly used for comparing the difference of gene expression among samples.

### Differential expression analysis

For each data set, and for each alignment and quantification protocol, we identified differentially expressed genes by using the edgeR, DESeq2, or DESeq packages.

### Quantification and statistical analysis

Statistical evaluations were conducted using R (version 4.3) or GraphPad Prism 9 (GraphPad Software, San Diego, CA). Comparative analysis between two groups was performed using the Student's t-test for parametric data or the Mann–Whitney U test for non-parametric data. For multi-group comparisons (≥3 groups), one-way ANOVA or the Kruskal–Wallis test was employed, depending on the data distribution. Results were considered statistically significant at an adjusted $p < 0.05$. Significance levels are denoted in figures as follows: *, $p < 0.05$; **, $p < 0.01$; ***, $p < 0.001$ and ****, $p < 0.0001$; "ns" represents non-significance. Detailed information regarding sample sizes (n), specific statistical tests, and biological replicates for each experiment is provided in the corresponding figure legends.

### Reporting summary

Further information on research design is available in the Nature Portfolio Reporting Summary linked to this article.

## Data availability

The data that support the findings of this study are available in the main text, the Supplementary Information, and Source Data file. The RNA-Seq data generated have been deposited in the National Center for Biotechnology Information (NCBI) Sequence Read Archive (SRA) database under the accession code PRJNA1239919 and PRJNA1337323. The metabolomics data to the OMIX repository (accession number: OMIX013390 and OMIX013372). Any additional information required to reanalyze the data reported in this work is available from the lead contact upon request. Source data are provided with this paper.

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

## Acknowledgements

Guangdong S&T Program (Z.H., 2024B1111150001); Shenzhen Medical Research Special Fund Project Target disease (Z.H., B2302036); National Key R&D Program of China (P.L., 2022YFA1304000); the program of Guangdong Provincial Clinical Research Center for Digestive Diseases (P.L., 2020B1111170004), National Key Clinical Discipline; National Natural Science Foundation of China (P.L., U21A20344; Z.H., 82273346); Science and Technology Program of Guangdong Province, China (Z.H., 2021B1212040017, J.W., 2024A04J4086); Key laboratory start-up project (Sixth Affiliated Hospital of Sun Yat-sen University) (Z.H., 2023WST03); Guangdong Medical Development Foundation (H.X., B2025247). This project was also supported by Guang Dong Cheung Kong Philanthropy Foundation.

## Author contributions

Z.H., P.L. and S.Q.J. supervised the study and designed the experiments. H.Y.Xie, J.Y.Y., J.J.W., W.H.M., H.Y.Xu, S.G., Y.C.X., Z.H.L., D.Y.L., M.J.C. and D.L.L. performed the experiments. H.Y.Xie and J.J.W. collected, analyzed and interpreted the data. H.Y.Xie and J.Y.Y. wrote the manuscript. Z.H., P.L. and S.Q.J. revised the manuscript. All authors read and approved the final version of the manuscript.

## Competing interests

The authors declare no competing interests.
