## [Transparent Peer Review file · Nature Communications]

Kynurenine mediates the chemotherapy-induced intestinal toxicity through modulation of gut microbiota

Corresponding Author: Dr Zhen He

Version 0:

Reviewer comments:

Reviewer #1

(Remarks to the Author)

Chemotherapy induced gastrointestinal toxicity is a major limiting factor in cancer treatment and understanding the mechanisms associated with this adverse side-effect has merit. This manuscript describes the development of oxaliplatin-induced gastrointestinal toxicity, specifically the breakdown of the gut epithelial barrier due to alteration in tryptophan metabolism. The authors show that patients treated with oxaliplatin develop GI toxicity that can be stratified as high or low toxicity that correlates with lower levels of white blood cells and neutrophils. The authors also note that there is significant upregulation of L-kynurenine in patients demonstrating higher toxicity. They then test the hypothesis that enhanced L-kynurenine is associated with GI toxicity by oxaliplatin in mice models. The gut toxicity is evaluated by body weight, clinical score, histology, spleen index, and expression of MUC2 and apoptosis across the various treatments. The studies in mice have the following findings: 1) Oxaliplatin causes significant GI toxicity correlating with increase in kynurenine in serum, 2) deletion of the metabolizing enzyme (IDO) reduces oxaliplatin-induced GI toxicity, 3) depletion of gut microbiome improves GI toxicity due to oxaliplatin in WT, but worsens in IDO knock out mice (with respect to spleen index, colon length, separation from muscularis mucosae and cell apoptosis; but not in clinical score: see below my comments on this). This suggests that the protective effect of IDO1 KO is dependent on the gut microbiome. The authors have further conducted fecal transplant to show that WT mice receiving microbiome from IDO1 KO donors protect against oxaliplatin induce GI toxicity and it is *L. johnsonii* that protects against GI toxicity. They conclude that lack of kynurenine metabolism allows for the increase in *L. johnsonii*. In a final series of experiments, the authors show that myeloid cells are the main site for L-kynurenine biosynthesis and that providing bacteria that degrade L-kynurenine protects against oxaliplatin – induced GI toxicity.

There are several major issues with this manuscript in its current form.

1) The experimental methodology is not provided in detail and thus it is difficult to evaluate the specific experimental paradigms.

There is a discrepancy in the time frame of OXA treatment and kynurenine measurement in mice. In Fig 1, the model is 3 OXA treatments over two weeks, while the rest of the studies are conducted with 2 OXA injections over one week. Thus, it remains unclear if KYN measurements reflect changes over the 2 weeks. What is meant by clinical score? There is no description of the scoring system or any reference to previous studies. What specific physical appearance and mobility parameters were determined? What is the spleen index? The histology images in figures S1 – S5 of the spleen are unclear. What is being shown with regard to the differences in spleen index.

There is no indication of how many males/females were used and no stratification based on sex. Some studies e.g. Figure 3 have only 4 mice (males or females?) and it is unclear how statistical analysis were carried out with this small numbers. One of the cardinal features of chemotherapy-induced gastrointestinal toxicity is the development of severe diarrhea. This has not been measured and neither are studies to show changes in intestinal permeability.

Throughout the manuscript it is stated that kynurenine was gavaged 3 time/week. I assume every alternate day?

2) There are some important controls missing from the overall study. IDO1 Knock out mice show improved GI toxicity, however the levels of tryptophan or kynurenine are not determined.

How effective is one week of ABX treatment in depleting gut bacteria. There is no difference in the clinical score between IDO1 K.O. and IDO KO + ABX while other parameters of GI toxicity are significantly altered. This needs to be addressed.

The studies on the microbiome analysis are confusing. Are both donor and recipient mice treated with OXA or just the recipient mice? If both groups are treated than a control of untreated microbiome is missing. This needs to be validated. Also

Figure 4E and F are not clear. In 4 what is the difference between WT and control (4E). If mice are treated with OXA in 4F, please explain why expression is higher in WT than control of 4E. Figure 4G needs better labeling. It is unclear which class is illustrated based on the color scheme.

If OXA treatment is only given to recipient mice, then there appears to be no effect of OXA on the microbiome and the only differences are in the genotype.

3) The conclusion of this study is unclear. It would help for the authors to provide a schematic illustration on the mechanism of how OXA alters GI toxicity. An important control missing is whether *L. johnsonii* shifts with OXA treatment in WT mice, resulting in increased expression of kynurenine and if this is prevented in IDO1^{fl/fl} yzCre mice.

Minor issues

The manuscript is not well written and some conclusions are difficult to follow.

The color scheme of the figures need to be better presented so that significant differences can be observed.

Fig 2K: Kynurenine structure is incorrect

Fig 3R: Should conduct an outlier test for the FMT-Ido^{-/-} MUC2 expression

Fig 5H: Unclear of the IDO^{-/-} vs IDO^{-/+}

Reviewer #2

(Remarks to the Author)

Xie et al. focus on mechanisms underlying chemotherapy-induced intestinal toxicity. They found changes in tryptophan metabolites and traced the toxicity (in their models) to IDO1-mediated production on kynurenine pathway (KP) products. KP products then altered the microbiota, driving intestinal damage. Although the manuscript reports some important observations, it has several problems of logic and lack of appreciation of alternative possibilities.

Specific comments:

Figure 1. In the first part of the figure, changes (relative) to metabolites are shown but from the plasma. KP pathway metabolites are generally detected in any scenario where a strong IFN response has happened or is ongoing – a good example is SARS-CoV-2 infection – as IFNs drive IDO1 expression. Thus, in this figure, the issues center on (i) the source of IFN and especially IFN-gamma in the patients and mouse model, (ii) the cells expressing IDO1, and most importantly, (iii) the concentration of KP pathway metabolites in the gut relative to the blood.

Figure 2. Following from Figure 1, the authors tie IDO1 to the toxicity (e.g., panel M-Q) using a whole animal IDO1 KO. However, these effects are partial (panels N, O). We still don't know who is making IDO1 at this stage. An orthogonal approach would be to start OXA treatment and then treat with epacadostat and/or lindrostat in both WT and IDO1 KO models. This would give two important pieces of information: (i) any "remnant" effects could plausibly be attributed to TDO2, and (ii) adaptation of the IDO1 KO to the loss of IDO1 from development would be countered.

Figure 3. I find the authors' interpretation of the data shown in B and C difficult to follow. The authors' conclusions are reported in lines 157-159 but are vague. Looking carefully at the data, the IDO1 KO and WT mice seem fundamentally different. Yet the authors focus on the "reversal" of the IDO1 "protective" effect (the red vs. orange symbols) but the WT ABX treated mice have the reverse phenotype. Thus, ABX treatment triggers a different outcome in the WT vs. KO mice, which is not properly explained. Bone marrow chimeras would probably help here as one set of variables can be controlled relative to the others.

Figure 4. This figure contains descriptive microbiota-related data. As this reviewer understands it, the absence of IDO1 creates a permissive environment for an altered microbiota that is protective (and does not happen in the WT, see supra). If true, then administration of IDO1 inhibitors should do the same in the WT (but not the KO). The authors don't explain how lower KP pathway metabolites control these effects or any molecular targets in host or microbiota. Overall, the section is confusing and incomplete. Other things can happen here – for example, KP pathway metabolites have been reported to control ferroptosis and stress signaling – this could occur in the intestinal epithelia (but less so in the IDO1 KO) and drive changes in microbiota.

Figure 5. Loss of enterocytes 2, 3: this could be via an effect such as loss of NRF2/AHR signaling (see, key papers from Opitz, Murray, etc.). We still have no information on the IDO1⁺ cells in this figure or the source of IFNs. It is not clear in the scRNAseq how the experiment was done – is this steady-state or with OXA, ABX, etc. The conclusion drawn (lines 231-232) is vague.

Figure 6. This line of investigation lacks a basis in logic given the issues raised above. Use of *E. coli* BL21 may be irrelevant. LysM-cre also deletes in neutrophils and can cause mosaic deletion in many hematopoietic lineages. In my view, the use of the Cre delete here (even though it has drawbacks) should have been used earlier. Lines 253-254 – "better therapeutic strategy" for what? Overall, this part of the paper was weak.

Reviewer #3

(Remarks to the Author)

Recent advances in cancer chemotherapy include the development of molecularly targeted agents and immune checkpoint inhibitors, which have raised expectations for reducing treatment-associated side effects. Nevertheless, traditional cytotoxic agents that non-selectively target proliferating cells continue to be used. These drugs often cause severe side effects, sometimes forcing treatment discontinuation, which remains a significant clinical issue. In this study, Xie and colleagues report that patients treated with oxaliplatin—a platinum-based chemotherapeutic agent—who experienced severe side effects exhibited elevated levels of L-kynurenine, a tryptophan metabolite. Their mouse model further demonstrated that oxaliplatin-induced intestinal toxicity was associated with increased L-kynurenine levels. Importantly, genetic ablation of IDO-1, an enzyme essential for L-kynurenine synthesis, mitigated this toxicity in mice, highlighting the key role of L-kynurenine in this process. Intriguingly, the attenuation of toxicity observed in *Ido1*-knockout mice was abolished by antibiotic treatment, suggesting a critical involvement of microbiota in mediating L-kynurenine's effects. The authors further identified *Lactobacillus johnsonii* as a key bacterial species whose reduction due to elevated L-kynurenine levels exacerbates oxaliplatin-induced intestinal toxicity in mice. These findings provide novel insights into the regulation of chemotherapy-associated side effects and represent a significant contribution to the field.

However, there are several important unresolved issues.

It is particularly disappointing that the authors did not investigate why elevated L-kynurenine levels lead to the reduction of *L. johnsonii*, and by what mechanism *L. johnsonii* confers protection against oxaliplatin-induced intestinal toxicity.

Specific concerns:

1. Previous research has already suggested that tryptophan metabolism is involved in responses to chemotherapy and that the gut microbiota contributes to gastrointestinal toxicity. Therefore, the finding that tryptophan metabolism modulates chemotherapy-induced intestinal toxicity via the microbiota is somewhat expected. Nonetheless, this study's identification of *Lactobacillus johnsonii* as a central microbial player is highly novel and noteworthy. Still, the most critical question—why elevated L-kynurenine reduces *L. johnsonii*, and how this bacterium mechanistically protects against toxicity—remains essentially unaddressed. The authors should explore these aspects in more depth.

2. It would also be interesting to know whether *Lactobacillus johnsonii* is reduced in patients treated with oxaliplatin who have higher levels of L-kynurenine.

3. Since L-kynurenine can also be synthesized via TDO (tryptophan 2,3-dioxygenase), its concentration in *Ido1*-knockout mice is unlikely to be zero. The authors should clarify the actual concentration of L-kynurenine in *Ido1*-KO mice. This would enable the identification of a threshold for sufficient L-kynurenine reduction to mitigate oxaliplatin-induced intestinal toxicity.

4. In lines 151–153, the authors state “The improvement of the weight loss, clinical scores, colon length and spleen index in *Ido1*-knockout mice were disappeared in *Ido1*-knockout mice after treatment with antibiotics.” However, in Figure 3C, the clinical scores appear similar between *Ido1*-KO mice treated with or without antibiotics. Why? Moreover, antibiotic treatment in WT mice improves clinical scores (Figure 3C), suggesting the existence of specific bacteria that exacerbate toxicity. This interpretation is further supported by Figure 3B, where antibiotic-treated WT mice appear to recover body weight. The authors should provide a clear explanation of these results.

5. In Figure 3P, fecal transplantation from *Ido1*-KO mice leads to lower Ki67 expression than transplantation from WT mice. The authors should clarify why this phenomenon occurs.

Version 1:

Reviewer comments:

Reviewer #1

(Remarks to the Author)

The authors have responded to most of my concerns. I still feel that the manuscript needs further improvement in the writing as there are several grammatical errors.

Reviewer #2

(Remarks to the Author)

The authors have done quite a bit of work for this manuscript and a number of improvements were made. Amazingly, despite being told the chemical structure of Kyn was wrong, the authors have now shown (Fig. 2K) that Kyn is shown as Top and the upper "Top" structure - which should be Kyn, is ALSO wrong (the amine group of the aryl ring is in the wrong place). Structures of Tryptophan and Kynurenine are available on the internet.

Reviewer #3

(Remarks to the Author)

The authors have provided extensive new data, corrected previous errors, and articulated a coherent mechanistic framework linking IDO1-dependent kynurenine production, microbiota dysbiosis, and intestinal toxicity. In my view, all major concerns that I raised in the initial review have been satisfactorily addressed, and the revised manuscript now represents a solid and well-substantiated contribution to the field. I therefore consider this manuscript suitable for publication in Nature Communications.

POINT-BY-POINT REPLIES TO THE REVIEWER COMMENTS

Dear Editor,

Thank you for the thorough and constructive feedback provided by the reviewers on our manuscript. We greatly appreciate the reviewers' insights and in response we have undertaken considerable additional work, which we believe has strengthened the manuscript. We enclose a point-by-point response to the reviewers' comments and have highlighted all changes to the manuscript in blue.

Reviewer #1

Chemotherapy induced gastrointestinal toxicity is a major limiting factor in cancer treatment and understanding the mechanisms associated with this adverse side-effect has merit. This manuscript describes the development of oxaliplatin-induced gastrointestinal toxicity, specifically the breakdown of the gut epithelial barrier due to alteration in tryptophan metabolism. The authors show that patients treated with oxaliplatin develop GI toxicity that can be stratified as high or low toxicity that correlates with lower levels of white blood cells and neutrophils. The authors also note that there is significant upregulation of L-kynurenine in patients demonstrating higher toxicity. They then test the hypothesis that enhanced L-kynurenine is associated with GI toxicity by oxaliplatin in mice models. The gut toxicity is evaluated by body weight, clinical score, histology, spleen index, and expression of MUC2 and apoptosis across the various treatments.

The studies in mice have the following findings: 1) Oxaliplatin causes significant GI toxicity correlating with increase in kynurenine in serum, 2) deletion of the metabolizing enzyme (IDO) reduces oxaliplatin-induced GI toxicity, 3) depletion of gut microbiome improves GI toxicity due to oxaliplatin in WT, but worsens in IDO knock out mice (with respect to spleen index, colon length, separation from muscularis mucosae and cell apoptosis; but not in clinical score: see below my comments on this). This suggests that the protective effect of IDO1 KO is dependent on the gut microbiome. The authors have further conducted fecal transplant to show that WT mice receiving microbiome from IDO1 KO donors protect against oxaliplatin induced GI toxicity and it is *L. johnsonii* that protects against GI toxicity. They conclude that lack of kynurenine metabolism allows for the increase in *L. johnsonii*. In a final series of experiments, the authors show that myeloid cells are the main site for L-kynurenine biosynthesis and that providing bacteria that degrade L-kynurenine protects against oxaliplatin – induced GI toxicity. There are several major issues with this manuscript in its current form.

Response: We sincerely thank the reviewer for the insightful comments and constructive feedback on our manuscript. We appreciate the time and effort dedicated to reviewing our work. In the following comments, we have clarified any changes made to our manuscript to address specific reviewer questions.

1) The experimental methodology is not provided in detail and thus it is difficult to evaluate the specific experimental paradigms.

There is a discrepancy in the time frame of OXA treatment and kynurenine measurement in mice. In Fig 1, the model is 3 OXA treatments over two weeks, while the rest of the studies are conducted with 2 OXA injections over one week. Thus, it remains unclear if KYN measurements reflect changes over the 2 weeks.

Response: The initial experiment employed a regimen of three intraperitoneal injections of

oxaliplatin (OXA; 20 mg/kg) administered once per week over two weeks. However, for subsequent experiments, the dosing regimen was standardized to two injections of OXA (20 mg/kg) administered once per week. This modification was implemented because administration of the third dose significantly increased mortality rates in our model, likely due to the cumulative exacerbation of chemotherapy-induced toxicity. In follow-up experiments, we longitudinally quantified serum and fecal L-kynurenine levels in the two-dose model. These results (Figure R1) demonstrate that L-kynurenine concentrations remain persistently elevated following the second OXA dose, thereby capturing the sustained biochemical changes relevant to the model's pathophysiology.

Figure R1. Serum levels of L-kynurenine of WT mice before and after oxaliplatin treatment.

What is meant by clinical score? There is no description of the scoring system or any reference to previous studies. What specific physical appearance and mobility parameters were determined? What is the spleen index? The histology images in figures S1 – S5 of the spleen are unclear. What is being shown with regard to the differences in spleen index.

Response: The clinical score was assessed using a validated composite grading system (supplementary Table 2), adapted from established models of chemotherapy-induced toxicity. This scoring system has been previously employed by our group in a similar manner to evaluate the severity of chemotherapy-related adverse effects (PMID: 38400752). The spleen index was calculated as: Spleen weight (mg) / Body weight (g). This metric quantitatively reflects splenic hypertrophy or atrophy, where increased values indicate immune activation and decreased values suggest immunosuppression. Histological differences in spleen sections (Figs S1-S5) demonstrate: white pulp atrophy in severe toxicity (PMID: 33122357). High-resolution images have been re-uploaded.

There is no indication of how many males/females were used and no stratification based on sex. Some studies e.g. Figure 3 have only 4 mice (males or females?) and it is unclear how statistical analysis were carried out with this small numbers.

Response: We thank the reviewer for highlighting these critical methodological issues. All studies used male C57BL/6 mice to control for sex-specific variables. Regarding sample sizes, we have now expanded all cohorts to ensure robust conclusions. Revised figures present in **Figure 6A-I**.

One of the cardinal features of chemotherapy-induced gastrointestinal toxicity is the development of severe diarrhea. This has not been measured and neither are studies to show changes in intestinal permeability.

Response: We appreciate this insightful comment. While diarrhea and changes in intestinal permeability are recognized features of chemotherapy-induced gastrointestinal toxicity, our study focused on a comprehensive evaluation of colonic length, intestinal edema and various intestinal pathological changes, such as Ki-67 (proliferation), MUC2 (goblet cell function), and TUNEL

(apoptosis). These established metrics provide robust evidence of intestinal damage and dysfunction, offering a detailed understanding of the structural and functional integrity of the gut barrier in response to chemotherapy. In future investigations, we will consider incorporating experimental assays of intestinal permeability (e.g., FITC-dextran assay) to more comprehensively evaluate chemotherapy-induced impairment of intestinal barrier function.

2) There are some important controls missing from the overall study. IDO1 Knock out mice show improved GI toxicity, however the levels of tryptophan or kynurenine are not determined.

Response: We thank the reviewer for this insightful suggestion. Our new data show that L-kynurenine levels in *Ido1*^{-/-} mice are significantly lower than in controls, both before and after oxaliplatin treatment (**Figure S2B**).

How effective is one week of ABX treatment in depleting gut bacteria. There is no difference in the clinical score between IDO1 K.O. and IDO KO + ABX while other parameters of GI toxicity are significantly altered. This needs to be addressed.

Response: We sincerely appreciate the reviewer's meticulous scrutiny. In response to the reviewer's question regarding the efficacy of our one-week antibiotic (ABX) regimen, we confirmed its effectiveness through bacterial culture of fecal samples collected from the mice after one week of receiving antibiotic-supplemented water. The results demonstrated a marked reduction in viable bacterial colonies, confirming that the treatment was effective in depleting the gut microbiota (**Figure R2**). They are correct in identifying an error in the original Figure 3C where the group labels for '*Ido1*^{-/-}' and 'WT+ABX' were inadvertently swapped. This occurred during figure assembly due to misaligned data identifiers in our analysis spreadsheet. Repeated experiments with new cohorts, and replaced all affected figures (**Figure 6A-I**).

Figure R2. Representative images of fecal bacterial cultures.

The studies on the microbiome analysis are confusing. Are both donor and recipient mice treated with OXA or just the recipient mice? If both groups are treated than a control of untreated microbiome is missing. This needs to be validated. Also Figure 4E and F are not clear. In 4 what is the difference between WT and control (4E). If mice are treated with OXA in 4F, please explain why expression is higher in WT than control of 4E. Figure 4G needs better labeling. It is unclear which class is illustrated based on the color scheme.

Response: We apologize for any confusion in the microbiome study description. Key clarifications: Donor mice received NO OXA treatment, FMT donors were healthy untreated mice. Recipient mice underwent OXA challenge post-FMT - as per **Figure 6A** workflow. Revised **Figures 7E** with explicit labeling and unified color scheme.

Thank you for your insightful comment regarding the observed differences in *Lactobacillus* abundance between our two independent animal experiments. Figure 4E compares the fecal abundance of *Lactobacillus* in wild-type mice following treatment with or without oxaliplatin.

Figure 4F compares the abundance of *Lactobacillus* in fecal samples from wild-type and *Ido1*^{-/-} mice following oxaliplatin treatment. We agree this point warrants careful consideration. The primary reason for such discrepancies often stems from batch effects during sample processing and analysis. Specifically, variations can arise from the fecal DNA extraction process, where subtle differences in lysis efficiency, DNA yield, and purity may occur across different extraction batches. Similarly, qPCR detection can be subject to batch effects due to slight variations in reagent lots, instrument calibration between different assay runs. These cumulative factors can significantly influence the final quantification of specific bacterial taxa. To minimize such batch effects and enhance the comparability of the data, we have subsequently switched from an absolute quantification method based on standard curves to a relative quantification approach using a reference gene for normalization (**Figure 7F-7H**).

If OXA treatment is only given to recipient mice, then there appears to be no effect of OXA on the microbiome and the only differences are in the genotype.

Response: That's an interesting point. While genotype certainly plays a significant role in shaping the gut microbiome, we've found that OXA treatment also has a demonstrable impact on the fecal microbiota in mice. Our previously published work provides direct evidence of OXA's influence on the murine gut microbiome, even when considering genotypic variations. This indicates that OXA's effects on the microbiome are independent of, or additive to, any genotypic differences among the mice. Therefore, it's crucial to consider both genotype and OXA treatment as contributing factors to changes observed in the gut microbiome of recipient mice.

3) The conclusion of this study is unclear. It would help for the authors to provide a schematic illustration on the mechanism of how OXA alters GI toxicity.

Response: We agree that a clear illustration of the mechanism would be beneficial. As per your suggestion, we've now added a schematic illustration detailing how oxaliplatin (OXA) is proposed to alter gastrointestinal (GI) toxicity in the revised manuscript (**Figure 9**). We believe this visual aid will significantly enhance the understanding of our proposed mechanism.

An important control missing is whether *L. johnsonii* shifts with OXA treatment in WT mice, resulting in increased expression of kynurenine and if this is prevented in *IDO*^{fl/fl} *lyz*^{Cre} mice.

Response: We have investigated the effects of OXA treatment on *L. johnsonii* of WT mice. Our data indicate that OXA treatment leads to a decrease in the abundance of *L. johnsonii* in the feces of WT mice (**Figure 7F**). We performed the suggested experiments in *IDO*^{fl/fl} *lyz*^{Cre} mice treated with OXA. Our analysis of these mice showed that while serum kynurenine levels did not significantly increase compared to control groups after OXA treatment (**Figure 5I and 5J**), the abundance of *L. johnsonii* in their feces significantly increased (**Figure 7H**).

Minor issues

The manuscript is not well written and some conclusions are difficult to follow.

The color scheme of the figures need to be better presented so that significant differences can be observed.

Response: We sincerely apologize if the manuscript's writing made some conclusions difficult to follow. We understand the importance of clear and concise communication. We have thoroughly reviewed the entire manuscript and made revisions to enhance its readability and logical flow. We also acknowledge your valuable comment on the color schemes of our figures and their impact on distinguishing significant differences. We agree that visual clarity is paramount for data

interpretation. We have therefore reviewed our figures and adjusted the color palettes in areas where visual differentiation was less clear, aiming to improve contrast and ensure that key differences are more readily observable. We've focused on enhancing the distinctiveness of particularly critical data points and groups. The manuscript has been restructured to ensure a more logical and rational flow.

Fig 2K: Kynurenine structure is incorrect

Response: Thank you for catching the error in the kynurenine structure. We apologize for this oversight. We have corrected the kynurenine structure in the revised manuscript to ensure accuracy.

Fig 3R: Should conduct an outlier test for the FMT-*Ido*^{-/-} MUC2 expression

Response: We appreciate your attention to detail in our statistical analysis. You are correct that our data for MUC2 expression in the FMT-*Ido*^{-/-} group contains an outlier. We have conducted an outlier test, use Grubb's test to identify and remove the outliers from the data. To ensure the robustness of our conclusions, we have now performed further calculations both with and without this outlier. Importantly, our analysis consistently shows significant differences between the groups in both scenarios, whether the outlier is included or excluded. This demonstrates that our findings regarding MUC2 expression are robust and not driven by a single data point.

Fig 5H: Unclear of the IDO ^{-/-} vs IDO ^{-/+}

Response: In single-cell sequencing, we opted to use littermate heterozygous (+/-) mice as the control group for the homozygous knockout mice. Our primary rationale for this approach was to minimize potential confounding variables arising from differences in genetic background, maternal effects, and early-life environmental exposures. By utilizing littermates from the same dam and housed under identical conditions, we aimed to control for a multitude of external factors that can significantly influence complex biological phenotypes, including the microbiome. Crucially, we also validated the protein expression of IDO1 in both heterozygous and homozygous knockout mice. Our Western blot results clearly show that heterozygous mice express IDO1 protein, whereas homozygous mice show no IDO1 expression (**Figure R3**).

Figure R3. Western blot analysis of IDO1 and β -Tubulin in samples from Mouse IDO^{-/-} and Mouse IDO^{-/+}.

Reviewer #2 (Remarks to the Author):

Xie et al. focus on mechanisms underlying chemotherapy-induced intestinal toxicity. They found changes in tryptophan metabolites and traced the toxicity (in their models) to IDO1-mediated production on kynurenine pathway (KP) products. KP products then altered the microbiota, driving intestinal damage. Although the manuscript reports some important observations, it has several problems of logic and lack of appreciation of alternative possibilities.

Response: Thank you very much for the careful review and valuable feedback on our manuscript. We greatly appreciate you raising important points regarding the study's logic and the lack of appreciation for alternative possibilities. We agree that a deeper exploration of the mechanisms and logical chain connecting our findings is essential. As such, in the detailed responses and revisions

that follow.

Specific comments:

Figure 1. In the first part of the figure, changes (relative) to metabolites are shown but from the plasma. KP pathway metabolites are generally detected in any scenario where a strong IFN response has happened or is ongoing – a good example is SARS-CoV-2 infection – as IFNs drive IDO1 expression. Thus, in this figure, the issues center on (i) the source of IFN and especially IFN-gamma in the patients and mouse model, (ii) the cells expressing IDO1, and most importantly, (iii) the concentration of KP pathway metabolites in the gut relative to the blood.

Response: Thank you for your insightful comments regarding Figure 1 and the critical role of the L-Kynurenine Pathway (KP) in our study. You've raised important points concerning the source of IFN- γ , the cells expressing IDO1, and the compartmentalization of KP metabolites. We appreciate you highlighting the relevance of IFN- γ in driving IDO1 expression, as seen in various inflammatory contexts. To address your specific concerns: we have indeed investigated the IFN- γ response in our mouse model. Our data show that OXA treatment leads to an increase in IFN- γ concentration in both the serum and fecal samples of mice (**Figure 3A**). This indicates a systemic immune activation. We've performed flow cytometric analysis to identify the cellular sources of IFN- γ in our mouse model. Our results (**Figure 3B**) demonstrate that oxaliplatin treatment specifically increased IFN- γ production by CD8⁺ T cells, but not by other immune cells or intestinal epithelial cells. This helps to pinpoint the cellular contributors to the observed IFN- γ elevation. To further substantiate the critical role of IFN- γ , we conducted *in vivo* neutralization experiments. Administration of anti-IFN- γ antibody prior to oxaliplatin treatment exhibited a significant alleviation of both systemic side effects and intestinal damage (**Figure 3C-K and Figure S2C**). Concurrently, this depletion of IFN- γ led to a marked reduction in IDO1 expression (**Figure S2D**), directly confirming the IFN- γ -dependent regulation of IDO1 *in vivo*. While a direct comparison of absolute L-kynurenine concentrations between gut contents and blood is challenging due to differing biological matrices and units, our finding is that the relative changes in fecal L-kynurenine levels parallel those in the blood (**Figure R4**).

Figure R4. L-kynurenine levels in feces from mice treated with oxaliplatin.

Figure 2. Following from Figure 1, the authors tie IDO1 to the toxicity (e.g., panel M-Q) using a whole animal IDO1 KO. However, these effects are partial (panels N, O). We still don't know who is making IDO1 at this stage. An orthogonal approach would be to start OXA treatment and then treat with epacadostat and/or lindrostat in both WT and IDO1 KO models. This would give two important pieces of information: (i) any "remnant" effects could plausibly be attributed to TDO2, and (ii) adaptation of the IDO1 KO to the loss of IDO1 from development would be countered.

Response: Thank you for your highly insightful comments regarding Figure 2, particularly

concerning the observed effects with whole-animal IDO1 KO and the crucial question of the cellular source of IDO1. We also appreciate your valuable suggestion for an orthogonal approach using inhibitors. Regarding Partial Effects: We acknowledge that while IDO1 deletion significantly ameliorates OXA-induced toxicity as shown, the effects are not a complete abrogation. This suggests that while IDO1 plays a crucial role, other pathways or compensatory mechanisms may also contribute to the overall toxicity. Your point about pinpointing the specific cellular source of IDO1 is critical. To address this, our study also utilizes the IDO1^{fl/fl} lyz^{Cre} mouse model to investigate the role of IDO1 specifically in myeloid cells. As discussed in the manuscript and supported by data in **Figure 5A-I**, these experiments allow us to dissect the contribution of myeloid-derived IDO1 to the observed toxicity. Our findings from this conditional knockout model indicate that myeloid cells are indeed a significant source of IDO1 contributing to OXA-induced effects in our context.

We appreciate your suggestion to utilize inhibitors for a more comprehensive understanding. We agree that a pharmacological approach offers valuable mechanistic insights, and to further strengthen our findings, we have now conducted experiments using an IDO1 inhibitor. The results from these experiments, presented in **Figure 8A-I**, these results demonstrate that pharmacological inhibition of IDO1 with Epacadostat replicates the protective effect seen with genetic ablation of IDO1. Regarding TDO2, our current data, particularly the observed undetectable expression of TDO2 in intestinal tissue (**Figure R5**), suggest that TDO2 may not play a primary role in the intestinal side effects we are investigating. We acknowledge that inhibitor studies could offer additional insights into the developmental adaptation of constitutive IDO1 KOs, and we consider this an excellent direction for future investigations.

Figure R5. Western blot analysis of IDO1 and TDO2 in intestinal samples from wildtype mice.

Figure 3. I find the authors' interpretation of the data shown in B and C difficult to follow. The authors conclusions are reported in lines 157-159 but are vague. Looking carefully at the data, the IDO1 KO and WT mice seem fundamentally different. Yet the authors focus on the "reversal" of the IDO1 "protective" effect (the red vs. orange symbols) but the WT ABX treated mice have the reverse phenotype. Thus. ABX treatment triggers a different outcome in the WT vs. KO mice, which is not properly explained. Bone marrow chimeras would probably help here as one set of variables can be control relative to the others.

Response: We sincerely appreciate the reviewer's meticulous scrutiny. They are correct in identifying an error in the original Figure 3C where the group labels for ' *Ido1*^{-/-}' and 'WT+ABX' were inadvertently swapped. This occurred during figure assembly due to misaligned data identifiers in our analysis spreadsheet. Repeated experiments with new cohorts, and replaced all affected figures (**Figure 6A-I**). We believe this correction resolves the discrepancies you noted, including the seemingly "reverse phenotype" in WT ABX-treated mice.

Figure 4. This figure contains descriptive microbiota-related data. As this reviewer understands it,

the absence of IDO1 creates a permissive environment for an altered microbiota that is protective (and does not happen in the WT, see supra). If true, then administration of IDO1 inhibitors should do the same in the WT (but not the KO). The authors don't explain how lower KP pathway metabolites control these effects or any molecular targets in host or microbiota. Overall, the section is confusing and incomplete. Other things can happen here – for example, KP pathway metabolites have been reported to control ferroptosis and stress signaling – this could occur in the intestinal epithelia (but less of in the IDO1 KO) and drive changes in microbiota.

Response: Thank you for your comprehensive feedback on Figure 4. Regarding your insightful point about IDO1 inhibitors, we agree. As detailed in our response to your comments on Figure 8A-I, we have indeed conducted experiments using an IDO1 inhibitor in WT mice. As mentioned in the reply above.

You highlighted the need for a clearer explanation of how lower KP pathway metabolites control these effects and the underlying molecular targets. This is a critical mechanistic question. To address this, we have now included data exploring the direct impact of L-Kynurenine on *L. johnsonii* growth. Specifically, our in vitro experiments demonstrate that kynurenine can directly inhibit the growth of *L. johnsonii* (**Figure S5E**). Following transcriptome sequencing analysis, treatment with L-kynurenine led to a significant downregulation of several genes essential for the survival and proliferation of *L. johnsonii*. Specifically, the expression of AcpP (fatty acid synthesis), LepB and LspA (signal peptidases), DltB (cell wall modification), and RpoD (RNA polymerase) was decreased (**Figure S5J**). These findings suggest that L-kynurenine may inhibit the growth of *L. johnsonii*, compromising its ability to colonize the gut.

In addition to KP pathway metabolites controlling ferroptosis and stress signaling in intestinal epithelia, their potential role in regulating epithelial cell apoptosis may also indirectly contribute to microbiota alterations. We agree that these are intriguing possibilities. Furthermore, we have now investigated the downstream mechanisms by which L-kynurenine-induced microbiota dysbiosis enhances intestinal side effects. Our findings suggest that this may occur through the activation of the TNF- α /JNK pathway, which promotes epithelial cell apoptosis. Moreover, we found that supplementation with *L. johnsonii* can inhibit this pathway-induced apoptosis and ameliorate chemotherapy-induced intestinal toxicity (**Figure S6 A-D**). This provides a more detailed molecular link between the microbiota alterations caused by kynurenine and the host's inflammatory response, contributing to the observed gastrointestinal toxicity.

Figure 5. Loss of enterocytes 2, 3: this could be via an effect such as loss of NRF2/AHR signaling (see, key papers from Opitz, Murray, etc.). We still have no information on the IDO1+ cells in this figure or the source of IFNs. It is not clear in the scRNAseq how the experiment was done – is this steady-state or with OXA, ABX, etc. The conclusion drawn (lines 231-232) is vague.

Response: We understand your concern about the lack of specific information on IDO1+ cells and IFN sources within the context of Figure 5. To clarify these aspects: IDO1+ Cells: Our investigations into the cellular source of IDO1, as discussed in our response to your comments on **Figure 2**, indicate that myeloid cells are a significant source of IDO1 in our model.

Source of IFNs: As detailed in our response to your comments on **Figure 1**, we have assessed IFN- γ levels. Our data demonstrate an increase in IFN- γ in both serum and fecal samples following OXA treatment (**Figure 3A**). This indicates a systemic immune activation. We've performed flow cytometric analysis to identify the cellular sources of IFN- γ in our mouse model. Our results (**Figure**

3B) demonstrate that oxaliplatin treatment specifically increased IFN- γ production by CD8⁺ T cells, but not by other immune cells or intestinal epithelial cells. This helps to pinpoint the cellular contributors to the observed IFN- γ elevation.

You correctly point out that the scRNAseq experimental setup and the conclusions drawn were vague. We apologize for this lack of clarity. To address this: scRNAseq experimental design: we have revised the methods section to explicitly state that the scRNAseq experiment was performed on intestinal samples from untreated littermate *Ido1*^{-/-} mice and *Ido1*^{+/-} mice. This ensures it's clear that these analyses represent a steady-state comparison between the genotypes without additional interventions. We've also added more detail to the figure legend to precisely define the groups analyzed in the scRNAseq.

Figure 6. This line of investigation lacks a basis in logic given the issues raised above. Use of *E. coli* BL21 may be irrelevant. LysM-cre also deletes in neutrophils and can cause mosaic deletion in many hematopoietic lineages. In my view, the use of the Cre delete here (even though it has drawbacks) should have been used earlier. Lines 253-254 – “better therapeutic strategy” for what? Overall, this part of the paper was weak.

Response: We acknowledge the concerns about the logical flow of this section as it was previously presented. We agree that a clearer progression of our findings would enhance the paper's overall coherence. To address this, we have significantly restructured the results section, adjusting the order in which data from different models (including the conditional knockout) are presented. We believe this new logical sequence better builds the case for the role of IDO1 and the subsequent therapeutic implications.

You raised a valid point regarding the relevance of *BL21-KAT* in our study. We would like to clarify that our use of this engineered strain was carefully considered: it serves as a model to demonstrate how a gut microbe can be designed to degrade L-kynurenine. This approach directly aligns with our proposed mechanism, in which elevated L-kynurenine contributes to gut toxicity (**Figure 8J-8U**). By showing that a selectively modified bacterium can reduce L-kynurenine levels, we hope to provide preliminary yet insightful evidence that the gut microbiota can be therapeutically targeted. We believe these findings help build a preclinical foundation for potential future interventions—such as engineered probiotics—that may alleviate chemotherapy-induced side effects through modulation of the kynurenine pathway. Furthermore, we have also conducted related animal experiments using the IDO1 inhibitor Epcadostat. Collectively, these results demonstrate that pharmacological inhibition of IDO1 replicates the protective effect observed with genetic ablation of IDO1 (**Figure 8A-8I**), underscoring the translational potential of targeting this pathway for clinical applications.

We appreciate your comment regarding the Lyz-Cre model. As Lyz-Cre drives recombination in myeloid cell lineages, which inherently include neutrophils and other hematopoietic cells, its use allows us to specifically target IDO1 in these key immune populations. We believe that by combining data from this myeloid-specific conditional knockout with our whole-animal IDO1 knockout data and pharmacological inhibition studies, we provide a robust and comprehensive picture of IDO1's role in the context of relevant immune cells.

Reviewer #3 (Remarks to the Author):

Recent advances in cancer chemotherapy include the development of molecularly targeted agents and immune checkpoint inhibitors, which have raised expectations for reducing treatment-associated side effects. Nevertheless, traditional cytotoxic agents that non-selectively target proliferating cells continue to be used. These drugs often cause severe side effects, sometimes forcing treatment discontinuation, which remains a significant clinical issue. In this study, Xie and colleagues report that patients treated with oxaliplatin—a platinum-based chemotherapeutic agent—who experienced severe side effects exhibited elevated levels of L-kynurenine, a tryptophan metabolite. Their mouse model further demonstrated that oxaliplatin-induced intestinal toxicity was associated with increased L-kynurenine levels. Importantly, genetic ablation of IDO-1, an enzyme essential for L-kynurenine synthesis, mitigated this toxicity in mice, highlighting the key role of L-kynurenine in this process. Intriguingly, the attenuation of toxicity observed in *Ido1*-knockout mice was abolished by antibiotic treatment, suggesting a critical involvement of microbiota in mediating L-kynurenine's effects. The authors further identified *Lactobacillus johnsonii* as a key bacterial species whose reduction due to elevated L-kynurenine levels exacerbates oxaliplatin-induced intestinal toxicity in mice. These findings provide novel insights into the regulation of chemotherapy-associated side effects and represent a significant contribution to the field.

However, there are several important unresolved issues.

It is particularly disappointing that the authors did not investigate why elevated L-kynurenine levels lead to the reduction of *L. johnsonii*, and by what mechanism *L. johnsonii* confers protection against oxaliplatin-induced intestinal toxicity.

Response: We thank the reviewer for their insightful comments and constructive feedback. We acknowledge that the raised questions are critical for a deeper understanding of our findings. In the following sections, we will address each of these points in detail.

Specific concerns:

1. Previous research has already suggested that tryptophan metabolism is involved in responses to chemotherapy and that the gut microbiota contributes to gastrointestinal toxicity. Therefore, the finding that tryptophan metabolism modulates chemotherapy-induced intestinal toxicity via the microbiota is somewhat expected. Nonetheless, this study's identification of *Lactobacillus johnsonii* as a central microbial player is highly novel and noteworthy. Still, the most critical question—why elevated L-kynurenine reduces *L. johnsonii*, and how this bacterium mechanistically protects against toxicity—remains essentially unaddressed. The authors should explore these aspects in more depth.

Response: Thank you for your constructive feedback and for acknowledging the novelty of identifying *Lactobacillus johnsonii* as a central microbial player in our study. We agree that while the general involvement of tryptophan metabolism and the gut microbiota in chemotherapy-induced gastrointestinal toxicity is recognized, elucidating the specific mechanistic links is crucial for advancing the field. To this end, we have conducted new experiments and significantly expanded our mechanistic investigation in the revised manuscript. Specifically, our *in vitro* experiments demonstrate that kynurenine can directly inhibit the growth of *L. johnsonii* (**Figure S5E**). Following transcriptome sequencing analysis, treatment with L-kynurenine led to a significant downregulation of several genes essential for the survival and proliferation of *L. johnsonii*. Specifically, the expression of *AcpP* (fatty acid synthesis), *LepB* and *LspA* (signal peptidases), *DltB* (cell wall

modification), and RpoD (RNA polymerase) was decreased (**Figure S5J**). This provides a direct mechanistic explanation, suggesting that the elevated L-kynurenine levels, resulting from increased host IDO1 activity, create an unfavorable gut environment that suppresses *L. johnsonii* proliferation. Mechanism of *L. johnsonii*'s protective action: We have further investigated the downstream consequences of kynurenine-induced dysbiosis, particularly focusing on the reduction of *L. johnsonii*. Our findings suggest that this may occur through the activation of the TNF- α /JNK pathway, which promotes epithelial cell apoptosis. Moreover, we found that supplementation with *L. johnsonii* can inhibit this pathway-induced apoptosis and ameliorate chemotherapy-induced intestinal toxicity (**Figure S6 A-D**). This suggests that *L. johnsonii* (or the balanced microbiota it contributes to) plays a role in counteracting this specific apoptosis signaling pathway, thereby conferring protection against intestinal toxicity.

2. It would also be interesting to know whether *Lactobacillus johnsonii* is reduced in patients treated with oxaliplatin who have higher levels of L-kynurenine.

Response: Thank you for your excellent question regarding whether elevated L-kynurenine levels correlate with reduced *L. johnsonii* in oxaliplatin-treated patients. This is indeed a crucial point for clinical translation. To address your query, we have now supplemented our study with data from a new clinical cohort, comprising 10 patients with low toxicity and 18 with high toxicity. We observed that serum L-kynurenine levels were significantly higher, and fecal *L. johnsonii* abundance was significantly lower, in the high-toxicity group compared to the low-toxicity group (**Figure 7I**). This clinical finding strongly corroborates the mechanistic insights observed in our mouse models, further supporting the inverse relationship between L-kynurenine and *L. johnsonii*. It also highlights the potential clinical relevance of this mechanism in chemotherapy-associated gastrointestinal toxicity.

3. Since L-kynurenine can also be synthesized via TDO (tryptophan 2,3-dioxygenase), its concentration in *Ido1*-knockout mice is unlikely to be zero. The authors should clarify the actual concentration of L-kynurenine in *Ido1*-KO mice. This would enable the identification of a threshold for sufficient L-kynurenine reduction to mitigate oxaliplatin-induced intestinal toxicity.

Response: To address this, we have quantified the L-kynurenine levels in the serum samples of our *Ido1*^{-/-} mice (**Figure S2B**). Our data show that while L-kynurenine levels are indeed significantly reduced in *Ido1*^{-/-} mice compared to wild-type controls, they are not completely absent, confirming your point about TDO.

4. In lines 151–153, the authors state “The improvement of the weight loss, clinical scores, colon length and spleen index in *Ido1*-knockout mice were disappeared in *Ido1*-knockout mice after treatment with antibiotics.”

However, in Figure 3C, the clinical scores appear similar between *Ido1*-KO mice treated with or without antibiotics. Why?

Moreover, antibiotic treatment in WT mice improves clinical scores (Figure 3C), suggesting the existence of specific bacteria that exacerbate toxicity. This interpretation is further supported by Figure 3B, where antibiotic-treated WT mice appear to recover body weight.

The authors should provide a clear explanation of these results.

Response: We sincerely appreciate the reviewer's meticulous scrutiny. They are correct in

identifying an error in the original Figure 3C where the group labels for '*Ido1*^{-/-}' and 'WT+ABX' were inadvertently swapped. This occurred during figure assembly due to misaligned data identifiers in our analysis spreadsheet. Repeated experiments with new cohorts, and replaced all affected figures (**Figure 6A-I**). We are confident that the updated figures will now provide a clear and consistent representation of our results, resolving the apparent contradictions and allowing for a straightforward interpretation of how antibiotic treatment impacts *Ido1*^{-/-} and WT mice in the context of oxaliplatin-induced toxicity. We believe this correction is crucial for the clarity and validity of our conclusions.

5. In Figure 3P, fecal transplantation from *Ido1*-KO mice leads to lower Ki67 expression than transplantation from WT mice. The authors should clarify why this phenomenon occurs.

Response: We thank the reviewer for identifying the error in the original Figure 3P (now **Figure 6P** in the revised manuscript) regarding Ki67 expression. The group labels were inadvertently swapped, creating a misleading impression. We have corrected this in the revised manuscript and apologize for the oversight.

POINT-BY-POINT REPLIES TO THE REVIEWER COMMENTS

Dear Editor,

Thank you very much for your email and for the good news regarding our manuscript. We are delighted to learn that the journal is happy, in principle, to publish our work under an open access license. We are pleased to accept this offer.

Thank you also for the thorough and constructive feedback provided by the reviewers. We enclose a point-by-point response to the reviewers' comments and have highlighted all changes to the manuscript in blue. We hope that these revisions satisfactorily address all concerns raised.

Reviewer #1

The authors have responded to most of my concerns. I still feel that the manuscript needs further improvement in the writing as there are several grammatical errors.

Response: We thank the reviewer for this final important comment regarding the writing quality and remaining grammatical errors. We sincerely apologize for this oversight in the previous revision. We agree that linguistic clarity is paramount for the manuscript's readability.

To address this concern comprehensively, we have commissioned a premium professional English language editing service to thoroughly review and polish the entire manuscript. This service was provided by an editing company affiliated with Nature Research, specializing in scientific and technical documentation.

Reviewer #2:

The authors have done quite a bit of work for this manuscript and a number of improvements were made. Amazingly, despite being told the chemical structure of Kyn was wrong, the authors have now shown (Fig. 2K) that Kyn is shown as Top and the upper "Top" structure - which should be Kyn, is ALSO wrong (the amine group of the aryl ring is in the wrong place). Structures of Tryptophan and Kynurenine are available on the internet.

Response: We sincerely thank the reviewer for their continued vigilance regarding the chemical structures. We are deeply sorry that the structural error for Kynurenine (Kyn) persisted in Figure 2K despite previous feedback, specifically concerning the incorrect placement of the amine group (-NH₂) on the aryl ring. We agree this is a critical error. To prevent any recurrence of such errors, all chemical structures throughout the figures have been redrawn using the Nature Chemistry template settings. This adherence to a professional, high-precision standard ensures that all compounds, including Kynurenine and Tryptophan, are chemically accurate and correctly represented. We thank the reviewer once again for identifying this persistent error and helping us maintain the chemical integrity of the manuscript.

Reviewer #3:

The authors have provided extensive new data, corrected previous errors, and articulated a coherent mechanistic framework linking IDO1-dependent kynurenine production, microbiota dysbiosis, and intestinal toxicity. In my view, all major concerns that I raised in the initial review have been satisfactorily addressed, and the revised manuscript now represents a solid and well-substantiated

contribution to the field. I therefore consider this manuscript suitable for publication in Nature Communications.

Response: We are exceptionally grateful for this final, highly positive assessment of our revised manuscript. We deeply appreciate the time and effort the reviewer dedicated to providing such thorough and constructive feedback throughout the entire review process.